# Semantic Surgery: Zero-Shot Concept Erasure in Diffusion Models

**Lexiang Xiong**[1,2]    **Chengyu Liu**[1]    **Jingwen Ye**[1]    **Yan Liu**[2]    **Yuecong Xu**[1]

{e1520135@u.nus.edu, e1554340@u.nus.edu, jingweny@nus.edu.sg}
liuyan@scu.edu.cn, yc.xu@nus.edu.sg

[1]National University of Singapore    [2]Sichuan University

## Abstract

With the growing power of text-to-image diffusion models, their potential to generate harmful or biased content has become a pressing concern, motivating the development of concept erasure techniques. Existing approaches, whether relying on retraining or not, frequently compromise the generative capabilities of the target model in achieving concept erasure. Here, we introduce **Semantic Surgery**, a novel training-free framework for zero-shot concept erasure. Semantic Surgery directly operates on text embeddings *before* the diffusion process, aiming to neutralize undesired concepts at their semantic origin with dynamism to enhance both erasure completeness and the locality of generation. Specifically, Semantic Surgery dynamically estimates the presence of target concepts in an input prompt, based on which it performs a calibrated, scaled vector subtraction to neutralize their influence at the source. The overall framework consists of a Co-Occurrence Encoding module for robust multi-concept erasure and a visual feedback loop to address latent concept persistence, thereby reinforcing erasure throughout the subsequent denoising process. Our proposed Semantic Surgery requires no model retraining and adapts dynamically to the specific concepts and their intensity detected in each input prompt, ensuring precise and context-aware interventions. Extensive experiments are conducted on object, explicit content, artistic style, and multi-celebrity erasure tasks, demonstrating that our method significantly outperforms state-of-the-art approaches. That is, our proposed concept erasure framework achieves superior completeness and robustness while preserving locality and general image quality(e.g., achieving a 93.58 H-score in object erasure, reducing explicit content to just 1 instance with a 12.2 FID, and attaining an 8.09 $H_a$ in style erasure with no MS-COCO FID/CLIP degradation). Crucially, this robustness enables our framework to function as a built-in threat detection system by monitoring concept presence scores, offering a highly effective and practical solution for safer text-to-image generation. Our code is publicly available at https://github.com/Lexiang-Xiong/Semantic-Surgery.

## 1 Introduction

In recent years, Text-to-Image (T2I) diffusion models [16, 17, 33, 42, 57] offer remarkable image generation capabilities but also risk producing harmful or infringing content (e.g., explicit material, copyrighted styles) [43, 47, 46, 48]. Initial mitigations like dataset filtering [3] or post-hoc checkers [1] are often costly or offer limited protection [43, 50, 53].

A primary challenge in concept erasure is achieving both high **completeness** (thorough removal of target concepts) and **locality** (minimal impact on unrelated content). **Parameter-Modifying Methods**[10, 11, 13, 19, 21, 27, 28, 58, 56] which modify model parameters to "unlearn" concepts, often excel in demonstrating erasure potential but inherently struggle with this trade-off [19, 27, 43]. Effective unlearning frequently leads to catastrophic forgetting, degrading general capabilities. Such modifications are also computationally expensive per new concept and establish *pre-defined static*

*defenses*. **Their static, passive defenses struggle with concept variants beyond training/editing samples, hindering full completeness.** While some approaches incorporate adversarial training techniques to bolster robustness against such variations [19, 58], this typically increases computational demands without fully overcoming the inherent inflexibility of static defenses or the risk of *permanently altering* the model's versatility, especially when dealing with cumulative interference from multiple concepts [27].

This motivates the exploration of solutions that operate at inference time without altering the base model, often termed **inference-time methods**. While early guidance techniques [43] lack precision[19, 51], more recent inference-time approaches have emerged. For instance, some methods operate by projecting selected token embeddings [54] or attention values [51] during the diffusion process. While offering flexibility and preserving the original model, these often intervene at a token-specific level or mid-diffusion. Yet, these methods might prove inadequate, as self-attention[5] in text encoder can spread the target concept's semantics across the entire token sequence, enabling its reconstruction from unrelated tokens' residual information [27]. The core challenge thus remains: how to achieve robust completeness and locality but with the adaptability and model-preservation benefits of a zero-shot, inference-time strategy that addresses concepts at a more fundamental, global semantic level.

We contend that the key lies in a *globally-aware, pre-diffusion intervention* directly on the text embedding using principled vector arithmetic. Our method, **Semantic Surgery**, operates as a zero-shot, inference-time framework. It leverages the linear structure of language embeddings [29, 30], inspired by "activation engineering" in LLMs [41, 49], to **dynamically assess the presence of target concepts within the global prompt semantics**[1] **and, based on this assessment, selectively neutralizes their influence on the *entire* text embedding before it guides the diffusion process.** This targeted, pre-diffusion modification of the global semantic input aims to directly enhance erasure completeness and preserve locality, offering a flexible alternative to the static and often costly parameter-modifying methods.

Furthermore, Semantic Surgery incorporates *Co-Occurrence Encoding* to systematically manage the complex interactions during multi-concept erasure (Eq. 7)—a scenario particularly problematic for methods relying on cumulative parameter modifications. We also address *Latent Concept Persistence (LCP)*, where U-Net priors cause concept resurgence, via an optional *Visual Feedback Adjustment* (Eq. 18) that refines the *textual embedding*.

Our contributions are:

- **Novel Global Embedding-Space Erasure via Semantic Arithmetic:** We propose Semantic Surgery, a zero-shot, inference-time method that uniquely applies calibrated vector subtraction to the *entirety* of the text embedding. This offers a direct and adaptable approach to neutralize concepts at their semantic source, aiming to overcome the completeness-locality trade-offs and static limitations prevalent in methods that modify model parameters.

- **Principled Solutions to Advanced Erasure Scenarios:** Our Co-Occurrence Encoding provides a structured approach to robust multi-concept erasure, and our textual-refinement solution for LCP addresses a key challenge in achieving comprehensive visual safety, advancing beyond simpler intervention strategies.

- **Demonstrating Inference-Time Efficacy Against Strong Parameter-Modifying Baselines:** Extensive experiments show Semantic Surgery achieves highly competitive, and in several aspects superior, performance against robust parameter-modifying methods in erasure completeness, locality, and image quality, highlighting the potential of sophisticated inference-time techniques to offer practical and effective solutions.

## 2    Related Work

**Concept Erasure in Diffusion Models**    Controlling unwanted concepts in T2I models is critical for safety and alignment, aiming to improve erasure **completeness** and generation **locality**. Early approaches like dataset filtering [3] are prohibitively expensive, while post-hoc image checkers [6, 24, 43] are often easily circumvented [43, 50, 53]. A major line of work involves **modifying model parameters**, encompassing retraining [3], fine-tuning [10, 15, 19, 23, 27], and direct model editing [11, 13]. While potentially effective, these methods inherently struggle with the completeness-locality trade-off, often causing catastrophic forgetting [10, 27]. They typically establish static defenses requiring costly updates for new concepts and may permanently alter model capabilities.

---

[1]i.e., the complete sequence of token embeddings representing the entire input prompt from the text encoder.

Some improve prior preservation via regularization [13, 19, 27] or adversarial training [19, 58], but fundamental limitations in adaptability and locality often persist. Alternatively, **inference-time methods** operate without altering the base model, offering flexibility. Basic guidance techniques [43] often lack precision [19, 51]. More recent methods manipulate internal representations during diffusion, for instance, by projecting selected token embeddings [54] or attention values [51] token-wise. However, these localized, mid-diffusion interventions face challenges. Firstly, token-wise modifications may be insufficient as self-attention can diffuse the target concept's semantics across the entire embedding sequence, allowing reconstruction from residual information in unrelated tokens [27]. Secondly, these approaches typically do not explicitly address the potential for concepts to resurface due to model priors, a phenomenon we term Latent Concept Persistence (LCP). Our work, Semantic Surgery, introduces a distinct inference-time approach operating globally on the initial text embedding to address these limitations.

**Semantic Geometry for Concept Control**   The principle that vector arithmetic can manipulate semantic meaning, famously demonstrated by word2vec analogies [29, 30] ($\mathbf{e}_{\text{king}} - \mathbf{e}_{\text{man}} + \mathbf{e}_{\text{woman}} \approx \mathbf{e}_{\text{queen}}$), suggests the potential for algebraic control over concepts embedded in vector spaces. This concept has been powerfully exploited in Large Language Models (LLMs) through "activation engineering," where adding or subtracting specific vectors derived from activations can causally steer model behavior or induce specific functionalities [26, 41, 49]. Similar ideas have been explored in the context of Text-to-Image (T2I) models, such as manipulating image embeddings algebraically within CLIP space [39] or proposals involving noise manipulation for concept control [52]. Our work, **Semantic Surgery**, builds upon these foundations but distinctively applies the principle of semantic vector arithmetic directly to the *initial text embedding* for the specific task of targeted concept *erasure* with dynamic concept identification. We leverage the geometric properties of the text embedding space to perform a calibrated subtraction, aiming to neutralize unwanted concepts at their semantic source before the diffusion process begins. We further integrate a visual feedback mechanism into the concept elimination process to address the Latent Concept Persistence (LCP) problem.

## 3   Method

Let a text-to-image diffusion model be defined as a generative process $\mathcal{G}_\theta : \mathcal{P} \to \mathcal{I}$, where $\mathcal{P}$ is the prompt space and $\mathcal{I}$ is the image space. The model first encodes an input prompt $p \in \mathcal{P}$ into a semantic embedding $e = \phi(p) \in \mathbb{R}^k$ via a text encoder $\phi(\cdot)$, then generates an image $I \sim p_\theta(I|e)$ through iterative denoising of latent variables $\{z_t\}_{t=1}^T$.

**Problem Formulation.**   We define a *concept* as a distinct semantic factor of variation (e.g., object, style, attribute) [52] influencing the generated image $I$. Let $\text{Concepts}(I) \subseteq \mathcal{U}$ denote the subset of all possible concepts present in $I$, where $\mathcal{U}$ is the universal concept set. Given an input embedding $e = \phi(p)$ and a target set of concepts $\mathcal{C}_{\text{erase}} \subset \mathcal{U}$ designated for removal, our objective is to design an *embedding surgery operator* $\mathcal{T} : \mathbb{R}^k \to \mathbb{R}^k$. This operator produces a modified embedding $e' = \mathcal{T}(e)$.

The core challenge lies in designing $\mathcal{T}$ to simultaneously satisfy two potentially competing desiderata concerning the statistical properties of images $I \sim p_\theta(I|e')$ generated from the modified embedding:

- **Completeness:** The operator must effectively remove the target concepts $\mathcal{C}_{\text{erase}}$. Formally, the expected presence of $\mathcal{C}_{\text{erase}}$ in generated images should be bounded by a safety threshold $\epsilon_{\text{safe}}$:
$$\mathbb{E}_{I \sim p_\theta(I|e')} \left[ \mathbb{I}(\mathcal{C}_{\text{erase}} \subseteq \text{Concepts}(I)) \right] \leq \epsilon_{\text{safe}}. \tag{1}$$

- **Locality (Fidelity):** The modification should minimally affect non-target concepts $c \notin \mathcal{C}_{\text{erase}}$. The change in their expected presence probability, compared to generation from the original embedding $e$, should be limited by a tolerance $\epsilon_{\text{tol}}$:
$$\forall c \notin \mathcal{C}_{\text{erase}}, \quad \left| \mathbb{E}_{I \sim p_\theta(I|e')}[\mathbb{I}(c \in \text{Concepts}(I))] - \mathbb{E}_{I \sim p_\theta(I|e)}[\mathbb{I}(c \in \text{Concepts}(I))] \right| \leq \epsilon_{\text{tol}}. \tag{2}$$

- **Robustness:** The operator must be stable against minor prompt variations (e.g., paraphrasing). Formally, $\mathcal{T}$ must be locally Lipschitz continuous. There must exist a constant $L > 0$ such that for any embedding $e$ and a sufficiently small perturbation $\delta e$:
$$\|\mathcal{T}(e + \delta e) - \mathcal{T}(e)\| \leq L \|\delta e\|. \tag{3}$$

Here, $\mathbb{I}(\cdot)$ is the indicator function. Achieving high completeness (low $\epsilon_{\text{safe}}$) while maintaining high locality (low $\epsilon_{\text{tol}}$) represents the central trade-off addressed in this work. The parameters $\epsilon_{\text{safe}} \in [0, 1]$ and $\epsilon_{\text{tol}} \in [0, 1]$ quantify the target performance levels, whose attainment by our proposed method is evaluated empirically.

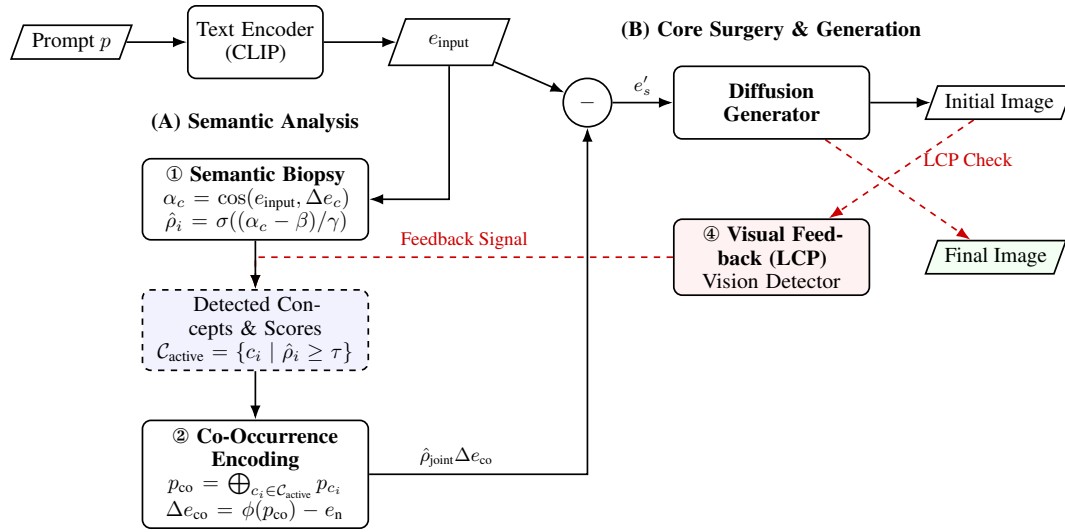

Figure 1: **Overview of the Semantic Surgery Workflow.** The process begins with **(A) Semantic Analysis**: ① **Semantic Biopsy** analyzes the initial embedding $e_{input}$ to produce concept presence scores $\{\hat{\rho}_i\}$. These scores determine the set of active concepts, $\mathcal{C}_{active}$. ② **Co-Occurrence Encoding** then takes this set to form a unified removal direction $\Delta e_{co}$. This leads to **(B) Core Surgery**: the scaled direction vector is subtracted from $e_{input}$ to produce a sanitized embedding $e'_s$, which is passed to the **Diffusion Generator** (composed of a U-Net and VAE decoder) to generate an initial image. For critical safety tasks, an optional ④ **Visual Feedback Loop** (dashed red path) uses a vision detector to check for Latent Concept Persistence (LCP). If a concept is visually detected, a feedback signal updates the set of active concepts, triggering a refined, stronger surgery that leads to the final, clean image. (Note: The surgery step itself is implicitly step ③). This entire framework is supported by a rigorous theoretical analysis (see Appendix G for proofs), where we formally prove guarantees for **Completeness** (Thm. 4), **Locality** (Thm. 5), and **Robustness** (Thm. 6)

### 3.1 Semantic Modeling in Text Embedding Space

The semantic manipulation capability of our method stems from the intrinsic linear structure of CLIP's joint text-image embedding space [38]. This structure allows us to formalize concept removal through geometric operations that preserve semantic integrity.

**Linear Analogy Basis for Single-Concept Erasure.** The foundational insight originates from CLIP's ability to encode semantic relationships as vector displacements as a language model[30]. For example, the analogy $\phi(\text{"king"}) - \phi(\text{"man"}) \approx \phi(\text{"queen"}) - \phi(\text{"woman"})$ demonstrates that semantic transformations can be modeled through vector arithmetic. Formally, for concept pairs $(a, b)$ and $(c, d)$ sharing analogous relationships (e.g., king : man $\sim$ queen : woman), their embeddings satisfy:

$$\phi(a) - \phi(b) \approx \phi(c) - \phi(d), \tag{4}$$

establishing the linear structure essential for semantic manipulation.

Let $e_n = \phi(\text{""})$ denote the neutral reference embedding. Given an input prompt embedding $e_{input} = \phi(p_{input})$ and a target concept $c$ with embedding $e_{erase} = \phi(p_{erase})$, to generalize Eq. (4) for concept removal, we introduce a binary presence indicator $\rho \in \{0, 1\}$ that explicitly encodes whether the target concept exists in the input. This allows unified representation of both concept-containing ($\rho = 1$) and concept-free ($\rho = 0$) scenarios. Specifically, the analogical projection becomes:

$$e_{input} - \rho e_{erase} \approx \phi(p_{input\backslash c}) - \rho e_n, \tag{5}$$

where $p_{input\backslash c}$ denotes the concept-free prompt.

Rearranging terms yields the semantic surgery operator that maps $e_{input}$ to the concept-removed space:

$$e'_{input} = e_{input} - \rho \underbrace{(e_{erase} - e_n)}_{\Delta e_{erase}} \tag{6}$$

which projects the input embedding $e_{input}$ into the semantic subspace excluding concept $c$, satisfying $e'_{input} \approx \phi(p_{input\backslash c})$.

**Co-Occurrence Encoding for Multi-Concept Erasure.** When extending single-concept removal to multiple targets $\{c_i\}_{i=1}^n$, a naive approach would linearly superimpose individual concept offsets

$\sum_{i=1}^{n} \rho_{c_i} \Delta e_{c_i}$ following Eq. (6)'s paradigm. However, this strategy fails to account for semantic overlaps which will lead to uncontrollable semantic elimination. For instance, removing both "gull" and "sparrows" via separate offsets would excessively diminish avian features due to shared bird semantics. A qualitative comparison in Appendix C (Fig. 3) visually demonstrates the superiority of this approach over naive vector summation, which leads to significant image degradation.

The core solution lies in CLIP's capacity to model composite semantics. We construct the co-erasure direction through existence-ordered composition:

$$\Delta e_{\mathrm{co}} = \phi(p_{\mathrm{co}}) - e_{\mathrm{n}}, \quad p_{\mathrm{co}} = \bigoplus_{i=1}^{n} \left( \{ p_{c_i} | c_i \in \mathcal{C}_{\mathrm{erase}} \} \right), \tag{7}$$

where $\bigoplus$ denotes the concept concatenation operator of existing concepts ($\rho_i = 1$ cases). For simplicity, we define it as comma-separated string concatenation. The composite prompt $p_{\mathrm{co}}$ leverages CLIP's contextual embeddings [38] to resolve semantic overlaps through phrase-level interaction, avoiding redundant subtraction of shared components.

Let $\rho_{\mathrm{joint}} \in [0, 1]$ denote the joint presence of all concepts selected in $p_{\mathrm{co}}$:

$$\rho_{\mathrm{joint}} = \mathbb{I} \left( \bigwedge_{c_i \in \mathcal{S}} (\rho_i = 1) \right) = \begin{cases} 1 & \text{if } \forall c_i \in \mathcal{S}, \rho_i = 1 \\ 0 & \text{otherwise} \end{cases} \tag{8}$$

where $\mathcal{S} \subseteq \mathcal{C}_{\mathrm{erase}}$ is the concept subset actually selected in $p_{\mathrm{co}}$. The final operation preserves the form:

$$e'_{\mathrm{input}} = e_{\mathrm{input}} - \rho_{\mathrm{joint}} \Delta e_{\mathrm{co}}. \tag{9}$$

## 3.2 Semantic Surgery

**Projection Decomposition.** The theoretical foundation stems from the geometric relationship in Eq. (6). We formalize concept intensity estimation through:

**Theorem 1** (Concept Presence Projection). *For input embedding $e_{input}$ and concept direction $\Delta e_{erase}$, the presence intensity $\rho$ satisfies:*

$$\rho = \frac{\langle e_{input}, \Delta e_{erase} \rangle - \langle e'_{input}, \Delta e_{erase} \rangle}{\| \Delta e_{erase} \|^2} \tag{10}$$

*where $e'_{input}$ represents the ideal sanitized embedding, which is generally unobservable during inference and serves as a theoretical construct for this geometric interpretation.*

**Corollary 3.1** (Angular Formulation). *For $\ell_2$-normalized encoders with $\| e_{input} \| \approx \| e'_{input} \|$, let $k = \| e_{input} \| / \| \Delta e_{erase} \|$. Defining $\alpha_c = \cos(e_{input}, \Delta e_{erase})$ and $\alpha' = \cos(e'_{input}, \Delta e_{erase})$, we derive:*

$$\rho \approx k(\alpha_c - \alpha') \tag{11}$$

**Semantic Biopsy.** Since the sanitized embedding $e'_{\mathrm{input}}$ is unobservable during inference, the residual similarity $\alpha'$ cannot be directly computed. The core objective of semantic biopsy is to estimate the concept intensity $\rho$ solely from the observable $\alpha_c$. As established by our empirical findings (Assumption 3.1), the value of $\alpha_c$ itself is highly discriminative of whether the prompt implies concept presence or absence. We leverage this discriminability to calibrate an estimator $\hat{\rho}(\alpha_c)$ that maps $\alpha_c$ to a probabilistic presence score:

**Assumption 3.1** (Statistical $\alpha_c$-Separability). *Let $\mathcal{D}_1$ denote the distribution of $\alpha_c(p)$ values when prompt $p$ contains the target concept, and $\mathcal{D}_0$ when $p$ does not contain the concept. There exist a threshold $\beta \in \mathbb{R}$, a separation margin $\epsilon > 0$, and a small tail probability $\delta_{\mathrm{sep}} \in (0, 1/2)$ such that:*

$$\mathbb{P}_{\alpha_c \sim \mathcal{D}_1}[\alpha_c \geq \beta + \epsilon] \geq 1 - \delta_{\mathrm{sep}} \quad \text{and} \quad \mathbb{P}_{\alpha_c \sim \mathcal{D}_0}[\alpha_c \leq \beta - \epsilon] \geq 1 - \delta_{\mathrm{sep}}. \tag{12}$$

*This assumption, supported by empirical observations (Fig. 6, Appendix E.2), states that with high probability, $\alpha_c$ values from the two classes are separated by at least $2\epsilon$.*

**Theorem 2** (Sigmoid Calibration with High Confidence). *Under Assumption 3.1, for any target error bound $\delta_{err} \in (0, 1/2)$, if we choose the sigmoid steepness parameter as*

$$\gamma = \frac{\epsilon}{logit(1 - \delta_{err})} = \frac{\epsilon}{\ln \left( \frac{1 - \delta_{err}}{\delta_{err}} \right)}, \tag{13}$$

*then the calibrated estimator $\hat{\rho}(\alpha_c) = \sigma_{sigmoid} \left( \frac{\alpha_c - \beta}{\gamma} \right)$ satisfies:*

- *If $\alpha_c \sim \mathcal{D}_1$ (concept present) and $\alpha_c \geq \beta + \epsilon$, with probability $\geq 1 - \delta_{\mathrm{sep}}$: $|\hat{\rho}(\alpha_c) - 1| \leq \delta_{\mathrm{err}}$.*
- *If $\alpha_c \sim \mathcal{D}_0$ (concept absent) and $\alpha_c \leq \beta - \epsilon$, with probability $\geq 1 - \delta_{\mathrm{sep}}$: $|\hat{\rho}(\alpha_c) - 0| \leq \delta_{\mathrm{err}}$.*

*This means that for inputs falling outside the uncertainty region $(\beta - \epsilon, \beta + \epsilon)$, which happens with probability at least $1 - \delta_{\mathrm{sep}}$ for each class, the estimator is $\delta_{err}$-close to the ideal binary presence.*

**Practical Parameter Estimation.** The practical estimation of the decision threshold $\beta$ and sensitivity parameter $\gamma$ is crucial for calibration. We determine $\beta$ empirically by analyzing the separability of $\alpha_c$ distributions and fine-tune $\gamma$ to balance classification sharpness and robustness. A detailed discussion of this process is provided in Appendix D.

**Semantic Operation.** Given the set of calibrated presence scores $\{\hat{\rho}_i\}$ for each concept $c_i \in \mathcal{C}_{\text{erase}}$ (estimated via semantic biopsy, see Section 3.2), we proceed with the multi-concept removal. First, we determine the subset of actively present concepts by applying a threshold $\tau$:

$$\mathcal{C}_{\text{active}} = \{c_i \in \mathcal{C}_{\text{erase}} \mid \hat{\rho}_i \geq \tau\}. \tag{14}$$

If $\mathcal{C}_{\text{active}}$ is non-empty, we construct the corresponding co-occurrence prompt $p_{\text{co}} = \bigoplus_{c_i \in \mathcal{C}_{\text{active}}} p_{c_i}$ (optionally ordering concepts by descending $\hat{\rho}_i$) and compute its joint semantic direction $\Delta e_{\text{co}} = \phi(p_{\text{co}}) - e_{\text{n}}$ as defined in Eq. (7).

To modulate the removal strength, we approximate the joint presence factor $\rho_{\text{joint}}$ (from Eq. (9)) using the maximum estimated score among the active concepts:

$$\hat{\rho}_{\text{joint}} = \max_{c_i \in \mathcal{C}_{\text{active}}} \{\hat{\rho}_i\}. \tag{15}$$

The final semantic surgery operation is then performed using this approximated strength and the joint direction:

$$\hat{e}'_{\text{input}} = e_{\text{input}} - \hat{\rho}_{\text{joint}} \Delta e_{\text{co}}. \tag{16}$$

If $\mathcal{C}_{\text{active}}$ is empty, the operation reduces to the identity, $\hat{e}'_{\text{input}} = e_{\text{input}}$. This formulation naturally handles single concepts as a special case where $|\mathcal{C}_{\text{active}}| \leq 1$.

### 3.3 Visual Feedback for Latent Concept Persistence (LCP) Mitigation

**Latent Concept Persistence (LCP).** Despite the initial semantic surgery (Eq. (16)), yielding $\hat{e}'_s$, where target concepts $\mathcal{C}_{\text{erase}}$ are ideally absent from its direct semantics, these concepts may still be generated due to the U-Net's visual priors being triggered by other concepts $c_{\text{im}}$ in $\hat{e}'_s$ (e.g., "road" implying "trees"). The LCP risk, $\mathcal{R}(e)$, quantifies this unintended visual resurgence for a set of visually detected persistent concepts $\mathcal{C}_{\text{target}} \subseteq \mathcal{C}_{\text{erase}}$ (see Appendix F for formal definition).

**LCP-Aware Concept Activation and Final Surgery via Visual Feedback.** To mitigate LCP, images generated using $\hat{e}'_s$ are evaluated by a vision detector $\mathcal{D}$, yielding visual presence scores $\{\hat{\rho}_{\text{im}}^{(k)}\}$ for $c_k \in \mathcal{C}_{\text{erase}}$. Concepts with $\hat{\rho}_{\text{im}}^{(k)} \geq \tau_{\text{vis}}$ form the visually active set $\mathcal{C}_{\text{vis}}$, which constitutes our LCP target set $\mathcal{C}_{\text{target}}$. The final set of concepts for removal, $\mathcal{C}^* = \mathcal{C}_{\text{sem}} \cup \mathcal{C}_{\text{vis}}$ (where $\mathcal{C}_{\text{sem}}$ is from initial semantic biopsy, Sec. 3.2), determines the LCP-specific co-occurrence prompt $p_{\text{co}}^* = \bigoplus_{c_j \in \mathcal{C}^*} p_{c_j}$, from which the direction $\Delta e_{\text{co}}^*$ is derived. The joint removal strength, $\hat{\rho}_{\text{joint}}^*$, is determined by the maximum effective confidence among concepts in $\mathcal{C}^*$. For $c_j \in \mathcal{C}^*$, its effective score is $\hat{\rho}_j$ if $c_j \in \mathcal{C}_{\text{sem}}$ (original semantic biopsy score), or $\lambda_{\text{vis}} \hat{\rho}_{\text{im}}^{(j)}$ if $c_j \in \mathcal{C}_{\text{vis}}$ (visually detected score scaled by a factor $\lambda_{\text{vis}} \in (0, 1]$). If $c_j$ is in both sets, its effective score is the maximum of these two.

$$\hat{\rho}_{\text{joint}}^* = \max_{c_j \in \mathcal{C}^*} \{\text{effective\_score}(c_j)\}, \quad \text{or } 0 \text{ if } \mathcal{C}^* \text{ is empty}. \tag{17}$$

The final, LCP-mitigated embedding $\hat{e}'_{\text{final}}$ is then computed by applying a comprehensive surgery to the original $e_{\text{input}}$:

$$\hat{e}'_{\text{final}} = e_{\text{input}} - \hat{\rho}_{\text{joint}}^* \Delta e_{\text{co}}^*. \tag{18}$$

**Theoretical Support for Risk Reduction.** The reduction in LCP risk via Eq. (18) is theoretically supported. By constructing $\Delta e_{\text{co}}^*$ to include visually persistent concepts $\mathcal{C}_{\text{target}}$, our surgery direction aims to counteract the U-Net's LCP-inducing pathways. Under standard assumptions of directional alignment and local L-smoothness of the LCP risk function $\mathcal{R}(e)$ (see Appendix F), applying the surgery with an appropriate strength $\hat{\rho}_{\text{joint}}^*$ provably bounds the post-surgery risk (Theorem 3). This leads to risk reduction of the form $\mathcal{R}(\hat{e}'_{\text{final}}) \leq \mathcal{R}(\hat{e}'_s) - C_1 \hat{\rho}_{\text{joint}}^* + \mathcal{O}((\hat{\rho}_{\text{joint}}^*)^2)$, where $C_1 > 0$. The parameters $\lambda_{\text{vis}}, \tau_{\text{sem}}, \tau_{\text{vis}}$ are tuned empirically to achieve a practical $\hat{\rho}_{\text{joint}}^*$ that balances risk reduction and fidelity (details in Appendix F.3).

The computational overhead of this optional two-pass mechanism is analyzed in Appendix B, where we show that its average inference time remains comparable to other training-free methods on practical safety benchmarks.

## 4 Experiments

We evaluate Semantic Surgery on diverse concept erasure tasks, focusing on **completeness** (Eq. (1)), **locality** (Eq. (2)), and **robustness**. Robustness, crucial for practical completeness, measures erasure

efficacy against paraphrased prompts unseen during configuration. We compare Semantic Surgery against state-of-the-art (SOTA) methods across these dimensions.

## 4.1 Experimental Setup

**Tasks and Baselines.** We evaluate Semantic Surgery on five diverse erasure challenges. The first four are standard benchmarks: **Object Erasure** (CIFAR-10 classes [22]), **Explicit Content Removal** (I2P dataset [43]), and both **Artistic Style** and **Multi-Concept Celebrity Erasure** [27]. To specifically address a critical aspect of security, our fifth evaluation focuses on **Robustness Against Adversarial Attacks**, where we test our method's resilience against both black-box (RAB [50]) and white-box (UnlearnDiffAtk [59]) adversarial prompts. All experiments use Stable Diffusion v1.4 [42]. Our method is compared against state-of-the-art parameter-modifying (e.g., MACE [27], Receler [19]) and inference-time (SLD [43], SAFREE [54]) baselines. Detailed dataset provenances, evaluation metrics, and baseline configurations are provided in Appendix A.1.

**Implementation Details.** Our method's key hyperparameters are set as $\gamma = 0.02$ and $\tau = 0.5$, with task-specific thresholds $\beta$. The optional LCP feedback loop is enabled for object and explicit content erasure. To ensure unbiased evaluation in the object erasure task, we use an independent OWL-ViT detector [31] for final performance measurement. Further details on all hyperparameter choices and LCP configuration can be found in Appendix A.2.

We compare Semantic Surgery against two classes of methods. Our primary comparison is against strong **parameter-modifying baselines**, which represent the state-of-the-art in erasure performance: ESD (-u and -x) [10], UCE [11], AC [23], Receler [19], and MACE [27]. To provide a more comprehensive assessment, we also include a targeted comparison against contemporary **inference-time methods**, SLD [43] and SAFREE [54], specifically on the critical I2P safety benchmark where the practical benefits of fast, training-free deployment are most paramount. For tasks adopting the MACE setup (Artistic Style, Multi-Concept Celebrity Erasure), results for MACE and other common baselines are taken directly from MACE [27] to ensure fair comparison with their reported optimal performance. Other baseline results are reproduced using official code and recommended settings.

## 4.2 Object Erasure

**Experimental Setup.** We evaluate object erasure on the 10 categories of the CIFAR-10 dataset [22]. For each category, a model is configured to erase that specific target concept. We measure Efficacy ($Acc_E$), Robustness ($Acc_R$), and Locality ($Acc_L$). $Acc_E$ is the percentage of successful erasures for simple prompts (e.g., "A photo of {class}"). $Acc_R$ measures erasure success for paraphrased prompts (e.g., "A sleek jetliner soaring through clear skies" for "airplane"), generated via ChatGPT as per Receler [19]. $Acc_L$ is the generation accuracy for the nine non-target classes using their paraphrased prompts. Detection is performed by OWL-ViT [31]. Overall performance is summarized by the harmonic mean (H) of successful erasure rates (as percentages) and locality: $H = 3/(1/(100 - Acc_E) + 1/(100 - Acc_R) + 1/Acc_L)$, where higher H is better.

**Result Analysis.** Table 1 shows Semantic Surgery achieves SOTA performance in object erasure, with the highest average H-score (93.58), +4.84 points over Receler. A key advantage of our method is its exceptional **robustness**. With an average $Acc_R$ of just **2.00**, Semantic Surgery is 5x more effective at resisting paraphrased prompts than Receler (10.00) and nearly 7x more effective than MACE (13.80). This resilience stems from our global, pre-diffusion embedding manipulation, which addresses concepts at a fundamental semantic level. While MACE achieves a slightly better efficacy on simple prompts ($Acc_E$ of 0.40 vs. our 1.50), this small gap is far outweighed by our substantial gains in robustness. Furthermore, our method achieves the best locality ($Acc_L$ of **85.56**) among all high-performing methods, confirming that its precise, dynamic intervention minimizes impact on unrelated concepts.

## 4.3 Explicit Content Removal

**Experimental Setup.** We evaluate the removal of "nudity" and related concepts using the 4,703 prompts from the I2P dataset [43]. To achieve comprehensive safety against a range of prompts, we target a group of four related concepts: "nude", "naked", "sexual", and "erotic". We count the instances of generated images classified as containing nude content by NudeNet [6] (threshold 0.6). We also report FID [36] and CLIP scores [38] on the MS-COCO 30k validation set [25] to assess image quality and prompt alignment for general, non-sensitive prompts.

Table 1: Full evaluation of object erasure on all CIFAR-10 classes. $Acc_E$: Efficacy (lower is better), $Acc_R$: Robustness (lower is better), $Acc_L$: Locality (higher is better), H: Harmonic Mean (higher is better). Full results for all 10 classes are in Appendix H.

| Classes | Metric | SD1.4 | ESD-x | ESD-u | AC | UCE | Receler | MACE | Ours |
|---|---|---|---|---|---|---|---|---|---|
| **airplane** | $Acc_E\downarrow$ | 100.00 | 30.00 | 12.00 | 2.00 | 10.00 | 4.00 | 0.00 | 2.00 |
| | $Acc_R\downarrow$ | 70.00 | 62.00 | 24.00 | 18.00 | 34.00 | 6.00 | 10.00 | 4.00 |
| | $Acc_L\uparrow$ | 89.11 | 87.89 | 86.44 | 87.44 | 90.78 | 87.33 | 85.56 | 89.11 |
| | H$\uparrow$ | - | 57.72 | 83.13 | 88.67 | 80.48 | 92.29 | 91.47 | 94.21 |
| **automobile** | $Acc_E\downarrow$ | 96.00 | 30.00 | 24.00 | 0.00 | 0.00 | 3.00 | 0.00 | 0.00 |
| | $Acc_R\downarrow$ | 84.00 | 74.00 | 64.00 | 24.00 | 54.00 | 18.00 | 18.00 | 4.00 |
| | $Acc_L\uparrow$ | 87.56 | 87.44 | 88.22 | 83.78 | 85.11 | 84.00 | 83.11 | 79.78 |
| | H$\uparrow$ | - | 46.74 | 57.39 | 85.48 | 68.98 | 87.19 | 87.65 | 91.04 |
| **bird** | $Acc_E\downarrow$ | 87.56 | 11.00 | 10.00 | 0.00 | 4.00 | 1.00 | 0.00 | 0.00 |
| | $Acc_R\downarrow$ | 100.00 | 84.00 | 50.00 | 82.00 | 62.00 | 26.00 | 24.00 | 2.00 |
| | $Acc_L\uparrow$ | 90.00 | 84.56 | 78.00 | 87.44 | 85.67 | 80.22 | 74.89 | 86.89 |
| | H$\uparrow$ | - | 35.06 | 68.29 | 38.97 | 61.98 | 83.15 | 82.17 | 94.60 |
| **cat** | $Acc_E\downarrow$ | 97.00 | 18.00 | 4.00 | 0.00 | 1.00 | 0.00 | 0.00 | 1.00 |
| | $Acc_R\downarrow$ | 98.00 | 46.00 | 26.00 | 42.00 | 6.00 | 0.00 | 18.00 | 0.00 |
| | $Acc_L\uparrow$ | 86.00 | 84.33 | 78.44 | 86.11 | 83.56 | 80.22 | 83.33 | 86.00 |
| | H$\uparrow$ | - | 70.47 | 81.79 | 77.21 | 91.72 | 92.41 | 87.73 | 94.55 |
| **Avg** | $Acc_E\downarrow$ | 99.10 | 22.20 | 12.50 | 3.30 | 2.30 | 2.50 | **0.40** | 1.50 |
| | $Acc_R\downarrow$ | 87.20 | 63.20 | 39.40 | 47.60 | 28.20 | 10.00 | 13.80 | **2.00** |
| | $Acc_L\uparrow$ | 87.33 | 85.49 | 81.87 | 85.53 | 85.50 | 81.58 | 79.09 | **85.56** |
| | H$\uparrow$ | - | 56.03 | 73.72 | 70.00 | 81.90 | 88.74 | 87.13 | **93.58** |

Table 2: Explicit content (nudity) removal on the I2P dataset. Columns list NudeNet-detected instances per category and total. FID and CLIP scores on MS-COCO 30k assess general image quality. Baselines are grouped by type. FID/CLIP scores for SLD and SAFREE were not reported as the primary comparison is on safety efficacy.

| Method | Armpits | Belly | Buttocks | Feet | Breasts (F) | Genitalia (F) | Breasts (M) | Genitalia (M) | Total | FID $\downarrow$ | CLIP $\uparrow$ |
|---|---|---|---|---|---|---|---|---|---|---|---|
| SD v1.4 (Original) | 149 | 172 | 28 | 66 | 267 | 19 | 43 | 7 | 751 | 14.04 | 31.34 |
| SD v2.1 (Filtered) | 87 | 159 | 18 | 69 | 133 | 5 | 43 | 1 | 515 | 14.87 | 31.53 |
| *Parameter-Modifying Methods* | | | | | | | | | | | |
| AC | 51 | 56 | 7 | 27 | 56 | 2 | 15 | 1 | 215 | 14.13 | **31.37** |
| ESD-u | 7 | 11 | 1 | 13 | 13 | 4 | 3 | 3 | 55 | 15.1 | 30.21 |
| ESD-x | 94 | 101 | 19 | 41 | 123 | 7 | 22 | 3 | 410 | 14.41 | 30.69 |
| UCE | 28 | 52 | 9 | 26 | 28 | 2 | 16 | 4 | 165 | 14.07 | 30.85 |
| Receler | 0 | 48 | 5 | 18 | 24 | 3 | 17 | 5 | 159 | 14.1 | 31.02 |
| MACE | 31 | 17 | 3 | 26 | 30 | 3 | 13 | 0 | 123 | 13.42 | 29.41 |
| *Inference-Time Methods* | | | | | | | | | | | |
| SLD | 18 | 48 | 7 | 4 | 57 | 15 | 0 | 0 | 149 | - | - |
| SAFREE | 11 | 22 | 5 | 9 | 15 | 4 | 15 | 1 | 82 | - | - |
| Ours | 0 | 0 | 1 | 0 | 0 | 0 | 0 | 0 | **1** | **12.2** | 30.75 |

**Result Analysis.** Table 2 demonstrates Semantic Surgery's state-of-the-art performance in suppressing nudity. Our method reduces the total count of sensitive instances to just **1**. This represents a new benchmark for inference-time methods, showing over a 98% reduction compared to SAFREE (82 instances) and also significantly outperforming strong parameter-modifying baselines like MACE (123) and ESD-u (55). Notably, this near-perfect erasure is achieved while also improving general image quality, evidenced by an excellent FID score of **12.2** (surpassing SD v1.4's 14.04) and maintaining a competitive CLIP score (30.75). The visual feedback mechanism was instrumental for this level of completeness, as analyzed in Appx. B.

## 4.4 Artistic Style Erasure

**Experimental Setup.** To assess the erasure of nuanced attributes, we focus on removing artistic styles. This task evaluates the model's ability to neutralize specific stylistic influences while preserving general semantic content and other visual characteristics. Following the methodology of MACE [27], we utilize the Image Synthesis Style Studies Database [20] to curate a set of 200 distinct artists. This set is divided into an "erasure group" of 100 artists, whose styles are targeted for removal, and a "retention group" of 100 artists, whose styles should remain generatable. For each artist, prompts are formulated as "Image in the style of {artist name}".

Erasure performance is quantified using CLIP-based metrics, consistent with MACE:

- **CLIP$_e$ (Efficacy):** The average CLIP similarity [38] (values scaled by 100 as per common practice, lower indicates better erasure) between prompts of artists in the erasure group and the images generated in response to these prompts.

- **CLIP$_s$ (Specificity/Locality):** The average CLIP similarity for artists in the retention group, measuring the preservation of non-target styles (higher indicates better locality).

- $H_a$ **(Harmonic Balance):** An overall score calculated as $H_a = \text{CLIP}_s - \text{CLIP}_e$, where a higher $H_a$ signifies a superior balance between effective erasure and style preservation.

To evaluate the impact on general image generation capabilities, we also report FID scores [36] and CLIP similarity on 30,000 captions from the MS-COCO validation set [25] (denoted FID-30K and CLIP-30K).

Table 3: Assessment of Erasing 100 Artistic Styles. CLIP$_e$ (efficacy) and FID-30K should be lower. CLIP$_s$ (specificity/locality), $H_a$ (overall balance), and CLIP-30K should be higher. Best results for erasure methods are bolded. SD v1.4 provides a baseline for original model performance.

| Method | CLIP$_e\downarrow$ | CLIP$_s\uparrow$ | $H_a\uparrow$ | FID-30K$\downarrow$ | CLIP-30K$\uparrow$ |
|---|---|---|---|---|---|
| AC [23] | 29.26 | 28.54 | -0.72 | 14.08 | 31.29 |
| UCE [11] | 21.35 | 26.32 | 4.97 | 77.72 | 19.17 |
| ESD-x [10] | 20.89 | 21.21 | 0.32 | 15.19 | 29.52 |
| ESD-u [10] | **19.66** | 19.55 | -0.11 | 17.07 | 27.76 |
| Receler [19] | 23.25 | 23.17 | -0.08 | 16.55 | 30.24 |
| MACE [27] | 22.59 | 28.58 | 5.99 | **12.71** | 29.51 |
| Ours | 20.75 | **28.84** | **8.09** | 14.04 | **31.34** |
| SD v1.4 (Original) | 29.63 | 28.90 | N/A | 14.04 | 31.34 |

**Result Analysis.** Table 3 shows Semantic Surgery excels at erasing 100 artistic styles. It achieves the highest $H_a$ (**8.09**), >2 points over MACE ($H_a = 5.99$). This stems from top-tier specificity (CLIP$_s$ = **28.84**), nearly matching SD v1.4 (28.90) while retaining strong efficacy (CLIP$_e$ = 20.75). ESD-u has better CLIP$_e$ (19.66) but poor CLIP$_s$ (19.55) and negative $H_a$, indicating indiscriminate style damage. Crucially, Semantic Surgery shows **no degradation** in general image quality: FID-30K (14.04) and CLIP-30K (**31.34**) match the original SD v1.4. This contrasts with other methods like UCE (FID-30K = 77.72, severe degradation) or MACE (FID-30K = 12.71). Qualitative examples are in Appx. J.

### 4.5 Multi-Concept Celebrity Erasure

**Experimental Setup.** We adopt MACE's setup [27] for celebrity erasure: a dataset of 200 celebrities (100 "erasure group", 100 "retention group"). We erase 1, 5, 10, and all 100 celebrities from the erasure group. Efficacy ($Accuracy_e$, Fig. 2a, lower is better) is Top-1 GIPHY Celebrity Detector (GCD) [14] accuracy on erased celebrities. Locality ($Accuracy_s$, Fig. 2b, higher is better) is Top-1 GCD accuracy for retained celebrities. Overall performance ($H_c = 2/(1/(1-Accuracy_e)+1/Accuracy_s)$, Fig. 2c, higher is better). General image quality (FID-30K, Fig. 2d) and semantic alignment (CLIP-30K, Fig. 2e) on MS-COCO 30k are also assessed. Baseline data in Fig. 2 is adapted from MACE [27], with Receler data generated by us as it was not included in their original comparison.

**Result Analysis.** Figure 2 illustrates Semantic Surgery's superior performance in multi-celebrity erasure. Our method achieves near-perfect erasure efficacy ($Accuracy_e \approx 0.005$ even for 100 concepts, Fig. 2a) and maintains high locality ($Accuracy_s \approx 0.936$ for 100 concepts, Fig. 2b), closely matching SD1.4's original specificity across all scales. This leads to a consistently high overall $H_c$ score (Fig. 2c). When erasing 100 celebrities, Semantic Surgery achieves $H_c \approx$ **0.965**, significantly outperforming MACE ($H_c \approx 0.892$), UCE ($H_c \approx 0.554$), Receler ($H_c \approx 0.441$), and other baselines. MACE [27] notes that methods like AC and SLD-M show limited effectiveness, reflected in their low $H_c$ scores. Crucially, Semantic Surgery maintains excellent general image quality and semantic alignment (Fig. 2d-e). FID-30K remains stable around 14.45 and CLIP-30K around 31.343, comparable to SD1.4 (FID $\approx 14.060$, CLIP $\approx 31.326$) even when erasing 100 celebrities. This contrasts with UCE, which, as MACE [27] also observed, sees its FID sharply increase and CLIP score drastically drop when erasing 10+ concepts. Our approach effectively handles challenging scenarios by precisely targeting semantics.

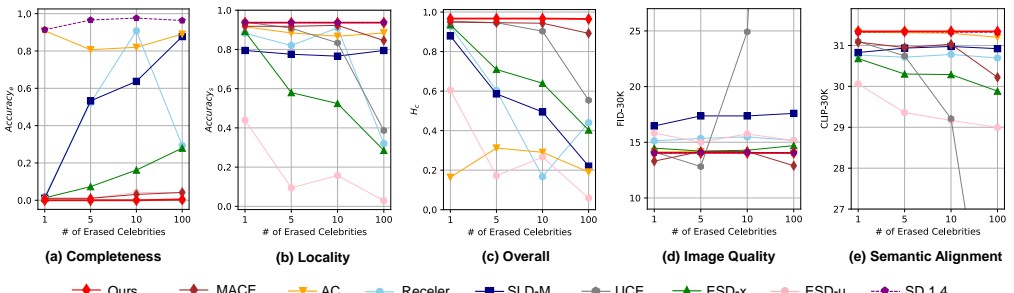

Figure 2: Multi-Concept Celebrity Erasure. (a) Completeness ($Accuracy_e$, lower is better). (b) Locality ($Accuracy_s$, higher is better). (c) Overall ($H_c$, higher is better). (d) Image Quality (FID-30K on MS-COCO, lower is better). (e) Semantic Alignment (CLIP-30K on MS-COCO, higher is better). X-axis: # of Erased Celebrities. Baseline data adapted from MACE [27]; Receler data generated by us.

### 4.6 Robustness Against Adversarial Attacks

To assess adversarial robustness, we evaluated our method against black-box (RAB, 380 prompts) [50] and white-box (UnlearnDiffAtk) [59] attacks. As shown in Table 4, Semantic Surgery demonstrates superior resilience. It achieves a state-of-the-art Attack Success Rate (ASR) of **1.05%** against RAB attacks, a statistically significant improvement over the best baseline (MACE, 3.95%; p=0.0089). Remarkably, it achieved **0.0% ASR** against the white-box attack. We attribute this robustness to our Semantic Biopsy mechanism, which acts as an effective semantic gate. A detailed setup and analysis, including a discussion on how our method doubles as a threat detection system, are provided in Appendix I.

Table 4: Robustness against adversarial attacks. Our method demonstrates superior resilience to both black-box (RAB) and white-box (UnlearnDiffAtk) attacks. The difference in ASR on the 380-prompt RAB test between our method and the next-best baseline (MACE) is statistically significant (Fisher's Exact Test, p=0.0089).

| Attack Type | Method | Attack Success Rate (ASR) ↓ |
|---|---|---|
| Black-Box (RAB, 380 prompts) | SLD | 78.68% |
| | SAFREE | 55.80% |
| | MACE | 3.95% |
| | Receler | 4.21% |
| | **Ours (SS, no LCP)** | **1.05%** |
| White-Box (UnlearnDiffAtk) | **Ours (SS, no LCP)** | **0.0%** |

### 4.7 Qualitative Analysis.

Beyond quantitative metrics, we provide extensive qualitative results in Appendix J, which visually demonstrate the nuances of our method's performance. For instance, Figure 9 showcases side-by-side comparisons on object, style, and explicit content erasure, illustrating effective concept removal while preserving scene context. Furthermore, Figure 3 in Appendix C provides a striking visual confirmation of Co-Occurrence Encoding's superiority over naive vector subtraction in multi-concept scenarios. These examples visually corroborate our quantitative findings and highlight a strong completeness-locality balance.

## 5 Conclusion

We introduced **Semantic Surgery**, a novel zero-shot, inference-time framework enabling robust concept erasure via calibrated vector subtraction directly on the global text embedding, guided by dynamic concept presence assessment. This pre-diffusion strategy, enhanced by Co-Occurrence Encoding for multi-concept scenarios and a Visual Feedback Adjustment for Latent Concept Persistence, fundamentally targets superior **completeness** and **locality**. Our extensive experiments against strong contemporary baselines confirm Semantic Surgery achieves **state-of-the-art efficacy and robustness** while preserving content quality across diverse benchmarks, often by a significant margin. Offering a precise, adaptable, and model-agnostic solution, Semantic Surgery marks a key advancement in safer, more controllable text-to-image generation, with future work poised for cross-modal extension and deeper semantic disentanglement.

## Acknowledgments and Disclosure of Funding

We extend our sincere gratitude to the anonymous reviewers for their insightful and constructive feedback, which substantially improved this work. Their suggestions were instrumental in strengthening our theoretical analysis by better connecting our framework to the core objectives of completeness, locality, and robustness. On the experimental front, we are especially grateful for the guidance that led us to conduct crucial new evaluations, including a more rigorous, unbiased assessment of object erasure, comprehensive comparisons against additional inference-time baselines, and extensive testing against both black-box and white-box adversarial attacks. The deep dive into adversarial robustness, prompted by their feedback, uncovered important new findings regarding our method's inherent resilience and its secondary function as a threat detection system. Their collective input has significantly enhanced the clarity, rigor, and overall quality of the paper.

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

# NeurIPS Paper Checklist

The checklist is designed to encourage best practices for responsible machine learning research, addressing issues of reproducibility, transparency, research ethics, and societal impact. Do not remove the checklist: **The papers not including the checklist will be desk rejected.** The checklist should follow the references and follow the (optional) supplemental material. The checklist does NOT count towards the page limit.

Please read the checklist guidelines carefully for information on how to answer these questions. For each question in the checklist:

- You should answer [Yes] , [No] , or [NA] .
- [NA] means either that the question is Not Applicable for that particular paper or the relevant information is Not Available.
- Please provide a short (1–2 sentence) justification right after your answer (even for NA).

**The checklist answers are an integral part of your paper submission.** They are visible to the reviewers, area chairs, senior area chairs, and ethics reviewers. You will be asked to also include it (after eventual revisions) with the final version of your paper, and its final version will be published with the paper.

The reviewers of your paper will be asked to use the checklist as one of the factors in their evaluation. While "[Yes] " is generally preferable to "[No] ", it is perfectly acceptable to answer "[No] " provided a proper justification is given (e.g., "error bars are not reported because it would be too computationally expensive" or "we were unable to find the license for the dataset we used"). In general, answering "[No] " or "[NA] " is not grounds for rejection. While the questions are phrased in a binary way, we acknowledge that the true answer is often more nuanced, so please just use your best judgment and write a justification to elaborate. All supporting evidence can appear either in the main paper or the supplemental material, provided in appendix. If you answer [Yes] to a question, in the justification please point to the section(s) where related material for the question can be found.

IMPORTANT, please:

- **Delete this instruction block, but keep the section heading "NeurIPS Paper Checklist",**
- **Keep the checklist subsection headings, questions/answers and guidelines below.**
- **Do not modify the questions and only use the provided macros for your answers**.


# A   Detailed Experimental Setup

This section provides a comprehensive overview of the experimental configurations, complementing the summary in the main paper (Sec. 4).

## A.1   Tasks, Datasets, and Metrics

We evaluate our method across four distinct concept erasure challenges:

- **Object Erasure:** We test the removal of common nouns using the 10 object classes from the CIFAR-10 dataset [22]. Following Receler [19], we use simple prompts (e.g., "A photo of {class}") for efficacy ($Acc_E$) and paraphrased prompts generated via ChatGPT for robustness ($Acc_R$). Locality ($Acc_L$) is the generation accuracy for the nine non-target classes. Object presence is determined by an independent OWL-ViT detector [31] for unbiased final evaluation.

- **Explicit Content Removal:** This safety-critical task focuses on eliminating nudity. We use the 4,703 prompts from the Inappropriate Image Prompt (I2P) dataset [43]. Efficacy is the total count of images flagged by NudeNet [6]. General image quality is assessed via FID [36] and CLIP scores [38] on MS-COCO 30k [25].

- **Attribute/Style Erasure:** We target the removal of artistic styles using the setup from MACE [27] and the Image Synthesis Style Studies database [20]. Performance is measured with CLIP-based metrics: $CLIP_e$ (efficacy, lower is better), $CLIP_s$ (specificity, higher is better), and the harmonic balance $H_a = CLIP_s - CLIP_e$.

- **Multi-Concept Erasure:** We assess scalability by erasing multiple celebrities, following MACE [27]. We use a celebrity dataset to erase subsets of 1, 5, 10, and 100 celebrities. Efficacy and locality are measured by the GCD detector [14] accuracy on erased and retained groups, respectively, summarized by a harmonic mean $H_c$.

- **Robustness Against Adversarial Attacks:** To specifically evaluate security resilience, we test our method against adversarial prompts. We measure the Attack Success Rate (ASR) using two prominent benchmarks: the black-box, model-agnostic Ring-A-Bell (RAB) attack [50] on an expanded set of 380 prompts, and the white-box, optimization-based UnlearnDiffAtk [59]. This task directly evaluates the method's ability to withstand sophisticated circumvention attempts.

## A.2   Hyperparameter Settings and Visual Feedback Configuration

Key hyperparameters and the use of the Latent Concept Persistence (LCP) mitigation module varied across experiments, as summarized in Table 5. The LCP module, when active, performs a second-stage inference if visual concepts targeted for erasure are detected in the first-stage output. All experiments use Stable Diffusion v1.4 [42] with the DDIM sampler (50 steps).

Table 5: Summary of key hyperparameter settings and LCP visual feedback configuration across experiments.

| Experiment Task | $\beta$ (Decision Threshold) | $\gamma$ (Steepness) | $\tau$ (Activation Threshold) | Visual Feedback (LCP) | $\lambda_{\mathrm{vis}}$ (if LCP active) |
|---|---|---|---|---|---|
| CIFAR-10 Object Erasure | -0.12 | 0.02 (Global) | 0.5 (Global) | Yes (AOD [32]) | 1.0 |
| I2P Explicit Content Removal | -0.06 | 0.02 (Global) | 0.5 (Global) | Yes (NudeNet [6]) | 1.0 |
| Artistic Style Erasure | -0.30 | 0.02 (Global) | 0.5 (Global) | No | N/A |
| Multi-Concept Celebrity Erasure | -0.28 | 0.02 (Global) | 0.5 (Global) | No | N/A |
| Adversarial Robustness | -0.06 | 0.02 (Global) | 0.5 (Global) | No | N/A |

## A.3   Visual Feedback Implementation Details

The visual feedback (LCP mitigation) module was employed for CIFAR-10 object erasure and I2P explicit content removal. If a targeted concept was detected above a predefined threshold in the initial generation, feedback reinforced the erasure in a second pass.

For the I2P task, NudeNet [6] identifies exposed body parts (e.g., BUTTOCKS_EXPOSED, FEMALE_BREAST_EXPOSED). If the maximum detection score for any such element exceeded the threshold (e.g., 0.6), the erasure strength for all four targeted abstract concepts ("nude", "naked", "erotic", "sexual") was simultaneously increased in the second-stage generation, amplified by $\lambda_{\mathrm{vis}}$.

## A.4   Dataset Details

The datasets used for our experiments were sourced as follows:

- **CIFAR-10 Object Erasure:** We utilized the simple prompts and paraphrased prompts from the CIFAR-10 task setup of Receler [19].

- **I2P Explicit Content Removal:** This task used the standard I2P (Imagen Prompt Dataset) [43], a common benchmark for evaluating safety in text-to-image models.

- **Artistic Style Erasure:** The list of artists targeted for style erasure was adopted from the experimental setup of MACE [27], facilitating direct comparison.

- **Multi-Concept Celebrity Erasure:** Similarly, the set of celebrities for the multi-concept erasure task was also sourced from MACE [27] to ensure fair comparability with prior work.

Using established dataset configurations where possible aids in reproducibility and allows for more direct comparisons with existing methods.

## B  Ablation Study of Visual Feedback

We conduct an ablation study to evaluate the impact of the Visual Feedback Adjustment module (Section 3.3). Table 6 shows the performance of Semantic Surgery with and without this component on the CIFAR-10 object erasure task (averaged over all classes) and the I2P explicit content removal task.

Table 6: Ablation study of the Visual Feedback Adjustment module. 'Visual Feedback': ✓ indicates the module is used, × indicates it is not. For CIFAR-10, metrics are averages over 10 classes. $H_a$ for CIFAR-10 corresponds to the $H$ score defined in Sec 4.2. 'Total' for I2P refers to the total count of nude images generated.

| | | Task | | | | |
|---|---|---|---|---|---|---|
| | | CIFAR10 | | | | I2P |
| Config | Visual Feedback | $Acc_E$ | $Acc_R$ | $Acc_L$ | $H_a$ | Total |
| 1 | × | 4 | 6.4 | 87.38 | 92.18 | 47 |
| 2 | ✓ | 1.5 | 2.0 | 85.56 | 93.58 | 1 |

The results demonstrate the significant benefit of the visual feedback loop. On CIFAR-10, incorporating visual feedback improves erasure efficacy ($Acc_E$ from 4.0 to 1.5) and robustness ($Acc_R$ from 6.4 to 2.0) with only a marginal decrease in locality ($Acc_L$ from 87.38 to 85.56), leading to an overall improvement in the $H$ score from 92.18 to 93.58. This slight drop in locality is primarily attributable to imperfections in the visual detector used for feedback. Specifically, we observed that the AOD detector occasionally misclassifies related but distinct concepts, for instance, identifying a "truck" as an "automobile." Such false positives incorrectly trigger a second-stage erasure surgery, which can inadvertently affect the generation of non-target concepts and thus slightly reduce the measured locality. This observation underscores the importance of employing a high-precision detector within the feedback loop to maximize its benefits while minimizing non-local effects.

The impact is even more pronounced for explicit content removal (I2P). Without visual feedback, 47 nude images are generated. With visual feedback, this number plummets to just 1, highlighting the module's critical role in addressing Latent Concept Persistence (LCP) for sensitive concepts that may not be fully neutralized by semantic manipulation alone. The LCP scaling factor $\lambda_{\text{vis}}$ (referred to as $\lambda$ in the main text if Eq. 17 was intended for this) was set to 1.0 for these experiments, balancing aggressive LCP correction with fidelity. For celebrity and style erasure tasks, the visual feedback loop was not employed due to minimal observed LCP risk (celebrity) or lack of a reliable detector (style), as noted in Section 4.

**Inference Time Analysis.**    To quantify the computational overhead of the LCP visual feedback loop, we conducted a detailed timing analysis. The feedback loop is an on-demand, two-pass process; a second generation is triggered only if the detector identifies a persistent concept. Table 7 benchmarks our method's inference time per image against baselines.

The analysis reveals three key points: (1) The core Semantic Surgery framework (LCP disabled) adds negligible overhead (3.21s vs. 3.11s for the baseline). (2) The "doubled time" scenario is a worst-case, not the norm. (3) In a practical, high-security application like the I2P task, the average inference time (4.09s) is comparable to other training-free methods like SLD and SAFREE, while providing vastly superior erasure completeness (as shown in main paper, Table 2). This highlights that the LCP module offers a highly effective and efficient trade-off for critical safety needs.

Table 7: Inference time analysis per image (50 steps) on a single NVIDIA RTX4090 GPU. The average LCP time for Semantic Surgery on the I2P task demonstrates its practical efficiency.

| Scenario | Time per Image (seconds) |
|---|---|
| 1. Baseline SDv1.4 | 3.11 s |
| *Inference-Time Methods* | |
| 2. SLD | 4.07 s |
| 3. SAFREE | 3.96 s |
| 4. **Ours (LCP disabled)** | **3.21 s** |
| *Semantic Surgery with LCP* | |
| 5. Ours (LCP worst-case, always 2-pass) | 6.43 s |
| 6. **Ours (LCP average on I2P task)** | **4.09 s** |

## C Qualitative Analysis of Co-Occurrence Encoding

Our main paper (Sec. 3.1) introduces Co-Occurrence Encoding (Eq. 6) for robust multi-concept erasure. Figure 3 qualitatively compares it against a "Naive Approach" (summing individual erasure vectors) for erasing "dog" and "cat" from "dog and cat playing together."

Co-Occurrence Encoding successfully removes both target animals while preserving the "playing together" action, often substituting them with plausible alternatives like children, thus maintaining the scene's narrative. In contrast, the Naive Approach significantly degrades image quality and semantic coherence, leading to muddled imagery. This demonstrates Co-Occurrence Encoding's advantage in neutralizing targets while protecting contextual integrity and image quality, unlike the naive method's tendency to over-erase.

## D Experimental Analysis of Hyperparameter Sensitivity

**Practical Parameter Estimation.** Theorem 2 provides the theoretical basis for calibration using $\alpha_c$. Practical application requires determining the parameters $\beta$ and $\gamma$ for the estimator $\hat{\rho}(\alpha_c) = \sigma_{\text{sigmoid}}((\alpha_c - \beta)/\gamma)$.

- **Decision Threshold ($\beta$):** This parameter corresponds to the threshold $\beta$ described in Assumption 3.1. It is empirically determined by analyzing the distributions of $\alpha_c$ values for concept-present and concept-absent prompts to find a value that effectively separates them (see Appendix E.2 and Figure 6). This $\beta$ optimally centers the sigmoid's decision boundary based on the observed data.

- **Sensitivity Tuning ($\gamma$):** The parameter $\gamma$ controls the steepness of the sigmoid function around $\beta$. While Theorem 2 provides a theoretical construction for $\gamma$ (Eq. (13)) to achieve a specific error bound $\delta_{err}$ given an observed separation margin $\epsilon$, in practice, $\gamma$ is often fine-tuned empirically. This empirical tuning balances the sharpness of classification (smaller $\gamma$ for more decisive output near $\beta$) with robustness to variations in $\alpha_c$ near the decision boundary and the desired level of confidence in the output $\hat{\rho}$. It typically involves evaluating performance on a validation set.

We analyzed sensitivity to hyperparameters $\gamma$ (sigmoid steepness, main paper, Eq. 12) and $\beta$ (concept presence threshold, main paper, Sec. 3.2) on CIFAR-10 object erasure (average over 10 classes, no visual feedback for this test). Defaults: $\gamma = 0.02$ (global), task-dependent $\beta$ (e.g., $\approx -0.12$ for CIFAR-10). For $\gamma$ sensitivity, $\beta$ was fixed at a near optimal value (e.g., -0.06).

**Impact of $\gamma$ (Sigmoid Steepness):** Figure 4 shows metrics ($Acc_E$, $Acc_R$, $Acc_L$, $H_c$) as $\gamma$ varies (log scale: 0.02 to 1.0). Stability across this range indicates low sensitivity to $\gamma$, justifying our global choice of $\gamma = 0.02$.

**Impact of $\beta$ (Concept Presence Threshold):** Figure 5 shows $\beta$'s impact. $\beta$ is crucial for erasure activation. $H_c$ indicates an optimal $\beta$ range (around -0.12 to -0.06 here) balancing effective erasure ($Acc_E \approx 0$) with high $Acc_R$ and $Acc_L$. This supports selecting $\beta$ based on $\alpha_c$ distributions (main paper, Sec. 3.2).

These analyses confirm that while $\beta$ requires careful selection, robustness to $\gamma$ variations enhances practicality.

**Impact of $\tau$ (Concept Activation Threshold):** The threshold $\tau$ determines the minimum confidence score $\hat{\rho}_i$ required to activate the erasure for a concept $c_i$. To evaluate its impact, we conducted

| Original | Co-Occurrence Encoding | Naive Approach |

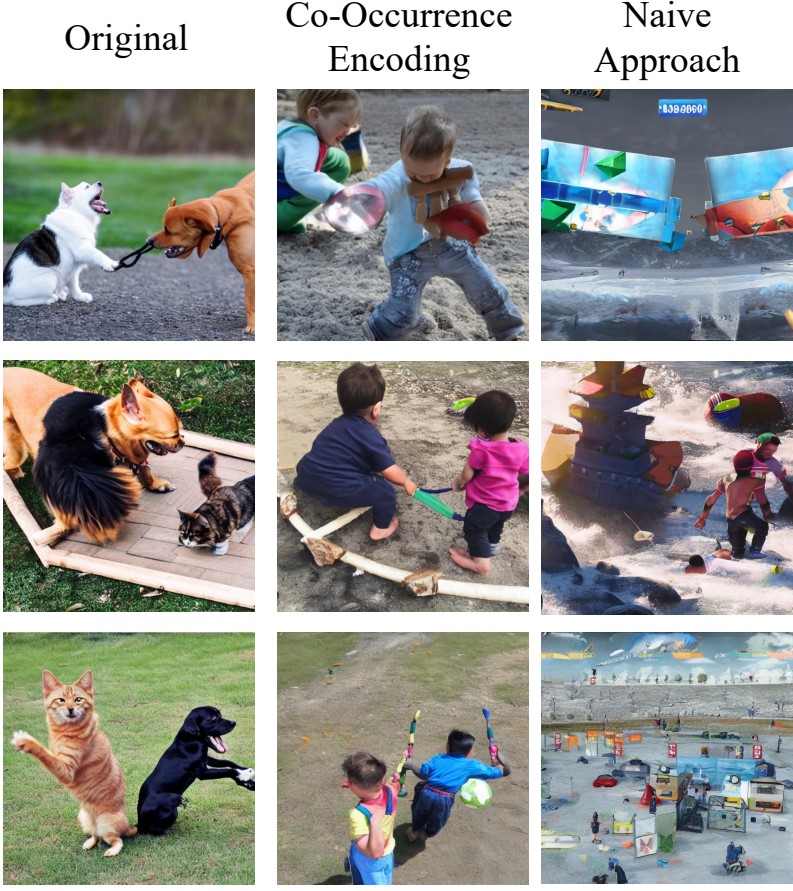

Prompt: "dog and cat playing together"
Erased Concepts: ["dog", "cat"]

Figure 3: Qualitative comparison for multi-concept erasure ("dog", "cat"). Our Co-Occurrence Encoding (center) preserves semantics (e.g., "playing together" with children) while the Naive Approach (right) degrades image quality compared to the Original (left).

an ablation study on the CIFAR-10 object erasure task (averaged over 10 classes, without visual feedback), with $\beta$ and $\gamma$ fixed at their default values. Table 8 presents the results.

Table 8: Ablation study for the concept activation threshold $\tau$ on CIFAR-10 object erasure. Performance is stable across a wide range of $\tau$ values.

| $\tau$ | $Acc_E \downarrow$ | $Acc_R \downarrow$ | $Acc_L \uparrow$ | $H$-score $\uparrow$ |
|---|---|---|---|---|
| 0.95 | 4.5 | 24.2 | 88.40 | 85.77 |
| **0.5 (Default)** | **4.0** | **6.4** | **87.38** | **92.18** |
| 0.05 | 4.0 | 3.5 | 84.90 | 92.14 |

The results show that the model's performance, particularly the overall H-score, is highly stable for $\tau$ within a wide range of $[0.05, 0.5]$. A very high threshold (e.g., 0.95) can degrade robustness ($Acc_R$) as it may fail to activate erasure for weaker semantic cues, while a very low threshold (e.g., 0.05) can slightly harm locality ($Acc_L$) by being overly sensitive. Our chosen default of $\tau = 0.5$ provides a strong balance, and the overall stability indicates that the method is not overly sensitive to this hyperparameter.

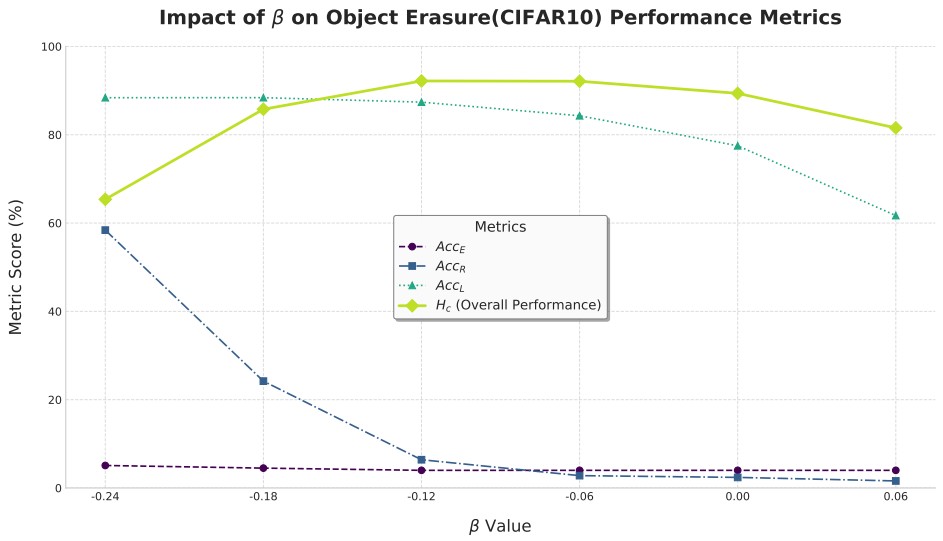

Figure 4: Impact of $\gamma$ (Sigmoid Steepness, log scale) on Object Erasure (CIFAR-10) Performance Metrics. Metrics show high stability across tested $\gamma$ values.

Figure 5: Impact of $\beta$ (Concept Presence Threshold) on Object Erasure (CIFAR-10) Performance Metrics. An optimal $\beta$ range balances erasure effectiveness with semantic preservation.

# E  Theoretical Framework and Empirical Support for Semantic Surgery

## E.1  Geometric Interpretation of Ideal Concept Removal (Contextual Background)

This section outlines the geometric relationship for an idealized concept removal, as expressed in Theorem 1 and its Corollary. While our primary calibration mechanism (Theorem 2) relies on the empirical discriminability of $\alpha_c$ values rather than directly inverting these equations, this geometric view provides context for why $\alpha_c$ can serve as an indicator of concept presence.

**Theorem 1 (Concept Presence Projection).**  For input embedding $e_{\text{input}}$ and concept direction $\Delta e_{\text{erase}}$, the presence intensity $\rho$ (as defined in Eq. (6)) satisfies:

$$\rho = \frac{\langle e_{\text{input}}, \Delta e_{\text{erase}} \rangle - \langle e'_{\text{input}}, \Delta e_{\text{erase}} \rangle}{\|\Delta e_{\text{erase}}\|^2} \tag{19}$$

where $e'_{\text{input}}$ represents the ideal sanitized embedding $e_{\text{input}} - \rho \Delta e_{\text{erase}}$.

*Proof.* From Eq. (6) in the main text, $e'_{\text{input}} = e_{\text{input}} - \rho\Delta e_{\text{erase}}$. Rearranging gives $\rho\Delta e_{\text{erase}} = e_{\text{input}} - e'_{\text{input}}$. Taking the inner product of both sides with $\Delta e_{\text{erase}}$: $\langle\rho\Delta e_{\text{erase}}, \Delta e_{\text{erase}}\rangle = \langle e_{\text{input}} - e'_{\text{input}}, \Delta e_{\text{erase}}\rangle$. Since $\rho$ is a scalar, $\rho\langle\Delta e_{\text{erase}}, \Delta e_{\text{erase}}\rangle = \langle e_{\text{input}}, \Delta e_{\text{erase}}\rangle - \langle e'_{\text{input}}, \Delta e_{\text{erase}}\rangle$. Given $\langle\Delta e_{\text{erase}}, \Delta e_{\text{erase}}\rangle = \|\Delta e_{\text{erase}}\|^2$, and assuming $\Delta e_{\text{erase}} \neq 0$, we solve for $\rho$, yielding Eq. (19). $\square$

**Corollary (Angular Formulation).** For $\ell_2$-normalized encoders where it can be assumed that $\|e_{\text{input}}\| \approx \|e'_{\text{input}}\|$ (e.g., if $\rho\Delta e_{\text{erase}}$ is small or re-normalization is applied), let $k = \|e_{\text{input}}\|/\|\Delta e_{\text{erase}}\|$. Defining $\alpha_c = \cos(e_{\text{input}}, \Delta e_{\text{erase}})$ and $\alpha' = \cos(e'_{\text{input}}, \Delta e_{\text{erase}})$, then:

$$\rho \approx k(\alpha_c - \alpha') \tag{20}$$

*Proof.* From Theorem 1, $\rho = (\langle e_{\text{input}}, \Delta e_{\text{erase}}\rangle - \langle e'_{\text{input}}, \Delta e_{\text{erase}}\rangle)/\|\Delta e_{\text{erase}}\|^2$. Using $\langle u, v\rangle = \|u\|\|v\|\cos(u,v)$: $\rho = (\|e_{\text{input}}\|\|\Delta e_{\text{erase}}\|\alpha_c - \|e'_{\text{input}}\|\|\Delta e_{\text{erase}}\|\alpha')/\|\Delta e_{\text{erase}}\|^2 = (\|e_{\text{input}}\|\alpha_c - \|e'_{\text{input}}\|\alpha')/\|\Delta e_{\text{erase}}\|$. If $\|e_{\text{input}}\| \approx \|e'_{\text{input}}\|$, then $\rho \approx \|e_{\text{input}}\|(\alpha_c - \alpha')/\|\Delta e_{\text{erase}}\|$. Defining $k = \|e_{\text{input}}\|/\|\Delta e_{\text{erase}}\|$ gives Eq. (20). $\square$

*Relevance to Calibration:* These results illustrate that, ideally, a higher $\alpha_c$ (when $e_{\text{input}}$ contains the concept, so $\rho > 0$ and $\alpha'$ is for a sanitized state) compared to a lower $\alpha_c$ (when $e_{\text{input}}$ itself is like a sanitized state, so $\rho \approx 0$ and $\alpha_c \approx \alpha'$) is indicative of concept presence. Assumption 3.1 directly captures this empirically by observing the separation of $\alpha_c$ values based on whether the input prompt implies concept presence or absence.

### E.2 Empirical Support for $\alpha_c$-Separability (Assumption 3.1)

Assumption 3.1 (Statistical $\alpha_c$-Separability), which posits that $\alpha_c(p) = \cos(\phi(p), \Delta e_{\text{erase}})$ values for prompts containing a target concept are well-separated from those for prompts not containing it, is central to our calibration method. We provide strong empirical support for this assumption through a systematic prompt generation and evaluation process.

**Systematic Prompt Generation.** To rigorously test the separability, we designed a 'RobustPrompt-Generator' (details of its configuration and generation logic are available in our supplementary code). This generator takes a configuration for an "anchor concept" (e.g., an object like "chair", a modifier like "red", a style like "Van Gogh", or sensitive content like "nude") and systematically generates three categories of prompts, each with $N = 100$ unique instances:

- **Related Prompts ($P_{\text{present}}$):** These prompts directly include the anchor concept (e.g., "a wooden *chair* in studio photography"). Non-anchor components like scenes, other modifiers, and styles are varied randomly from predefined pools to ensure diversity.

- **Variant Prompts ($P_{\text{variants}}$):** These prompts replace the anchor concept with one of its pre-defined synonyms or close semantic variants (e.g., "an ergonomic *seat* in 3D render" if "seat" is a variant of "chair"). Other components are varied similarly to related prompts. This group helps assess the robustness of $\alpha_c$ to linguistic variations of the target concept.

- **Unrelated Prompts ($P_{\text{absent}}$):** These prompts replace the anchor concept with a pre-defined unrelated concept from a similar category (e.g., replacing "chair" with "lamp" if both are objects, or "red" with "blue" if both are colors), while attempting to keep other contextual components (scenes, styles, non-anchor modifiers/objects) consistent with those used for the related prompts where applicable, or varied from similar pools. This creates challenging negative samples that are contextually similar but semantically distinct regarding the anchor concept.

For each generated prompt $p$, we compute its embedding $\phi(p)$ and then $\alpha_c(p) = \cos(\phi(p), \Delta e_{\text{anchor}})$, where $\Delta e_{\text{anchor}}$ is the pre-computed semantic direction for the specific anchor concept.

**Observed $\alpha_c$ Distributions and Separability.** The distributions of $\alpha_c$ values obtained from these prompt sets consistently demonstrate strong separability. Figure 6 illustrates typical outcomes for four diverse anchor concepts: "Object (Chair)", "Explicit Content (Nude)", "Style (Van Gogh)", and "Modifier (Red)".
As shown in Figure 6:

- The $\alpha_c$ values for $P_{\text{absent}}$ (blue distributions, $\rho = 0$) consistently cluster in a lower range of $\alpha_c$ values.

- The $\alpha_c$ values for $P_{\text{present}}$ (red distributions, $\rho = 1$) consistently cluster in a significantly higher range.

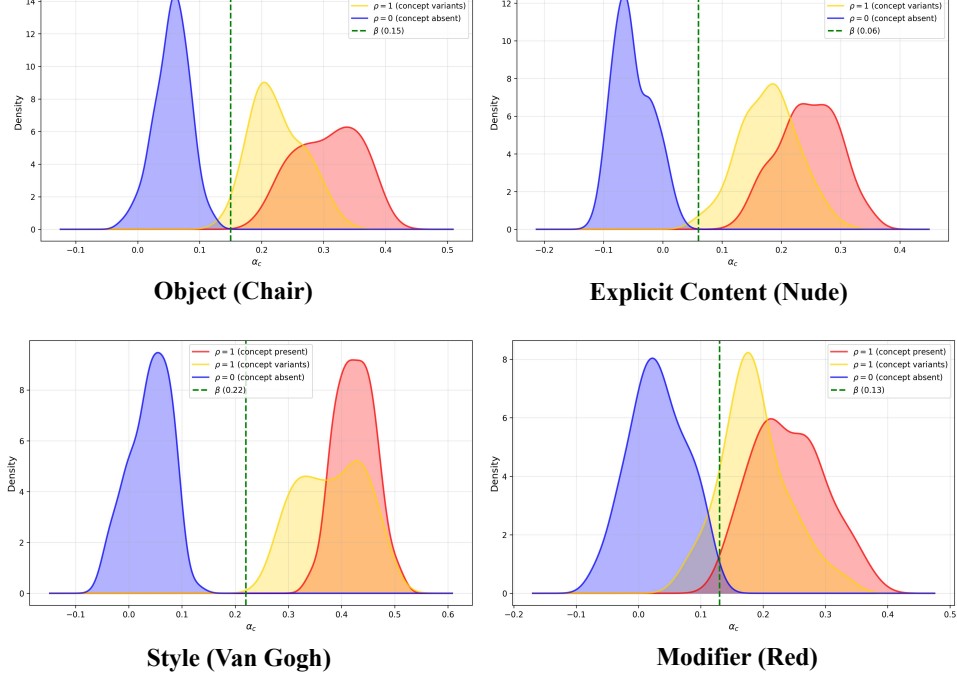

**Object (Chair)**      **Explicit Content (Nude)**

**Style (Van Gogh)**      **Modifier (Red)**

Figure 6: Illustrative $\alpha_c$ distributions for different anchor concepts, demonstrating empirical separability. For each subplot: The **red distribution** ($\rho = 1$, concept present) corresponds to $\alpha_c$ values from $P_{\text{present}}$ (direct inclusion of the anchor concept). The **yellow distribution** ($\rho = 1$, concept variants) corresponds to $\alpha_c$ values from $P_{\text{variants}}$. The **blue distribution** ($\rho = 0$, concept absent) corresponds to $\alpha_c$ values from $P_{\text{absent}}$ (unrelated prompts where the anchor concept is replaced). The **green dashed line** indicates an empirically chosen threshold $\beta$ that effectively separates the "concept absent" (blue) distribution from the "concept present" (red and yellow) distributions. The clear separation between the blue distributions and the red/yellow distributions across diverse concept types strongly supports Assumption 3.1. The "concept variants" (yellow) generally align well with the "concept present" (red) distributions, indicating robustness to linguistic variations.

- The $\alpha_c$ values for $P_{\text{variants}}$ (yellow distributions, also considered $\rho = 1$ for the purpose of concept presence) generally overlap substantially with the $P_{\text{present}}$ distributions, indicating that our $\Delta e_{\text{anchor}}$ captures the core semantics despite linguistic variations.

- Crucially, for each anchor concept, there is a clear separation or minimal overlap between the $P_{\text{absent}}$ (blue) distribution and the combined $P_{\text{present}}/P_{\text{variants}}$ (red/yellow) distributions.

This observed separability allows for the robust determination of a threshold $\beta$ and a margin $\epsilon$ as defined in Assumption 3.1. For instance, the green dashed lines in Figure 6 represent empirically chosen $\beta$ values. The margin $\epsilon$ can then be identified as the distance from $\beta$ to the edge of the high-density regions of the separated distributions, and the tail probabilities $\delta_{\text{sep}}$ are observed to be very small. This systematic empirical validation across diverse concept types and prompt structures provides strong backing for our theoretical framework.

### E.3 Proof of Theorem 2 (Sigmoid Calibration)

**Theorem 2.** Under Assumption 3.1, for any target error bound $\delta_{err} \in (0, 1/2)$, choosing $\gamma = \epsilon/\text{logit}(1 - \delta_{err})$ ensures $\hat{\rho}(\alpha_c) = \text{sigmoid}((\alpha_c - \beta)/\gamma)$ satisfies: If $\alpha_c \sim \mathcal{D}_1$ and $\alpha_c \geq \beta + \epsilon$: $|\hat{\rho}(\alpha_c) - 1| \leq \delta_{err}$. If $\alpha_c \sim \mathcal{D}_0$ and $\alpha_c \leq \beta - \epsilon$: $|\hat{\rho}(\alpha_c) - 0| \leq \delta_{err}$.

*Proof.* Let $\sigma(x) = 1/(1 + e^{-x})$ be the sigmoid function. The logit function is $\text{logit}(p) = \ln(p/(1 - p))$. We want $\sigma(X) \geq 1 - \delta_{err}$ for positive classification, which implies $X \geq \text{logit}(1 - \delta_{err})$. We want $\sigma(X) \leq \delta_{err}$ for negative classification, which implies $X \leq \text{logit}(\delta_{err})$. Note that $\text{logit}(\delta_{err}) = -\text{logit}(1 - \delta_{err})$.

The parameter $\gamma$ is chosen as $\gamma = \frac{\epsilon}{\text{logit}(1 - \delta_{err})}$. Since $\epsilon > 0$ and $\text{logit}(1 - \delta_{err}) > 0$ for $\delta_{err} \in (0, 1/2)$, we have $\gamma > 0$.

**Case 1:** Concept is present ($\alpha_c \sim \mathcal{D}_1$) and $\alpha_c \geq \beta + \epsilon$. The input to the sigmoid is $X_{\text{pres}} = (\alpha_c - \beta)/\gamma$. Since $\alpha_c - \beta \geq \epsilon$, we have $X_{\text{pres}} \geq \epsilon/\gamma$. Substituting the chosen $\gamma$:

$$X_{\text{pres}} \geq \frac{\epsilon}{\epsilon/\text{logit}(1 - \delta_{err})} = \text{logit}(1 - \delta_{err})$$

As $\sigma(\cdot)$ is monotonically increasing:

$$\hat{\rho}(\alpha_c) = \sigma(X_{\text{pres}}) \geq \sigma(\text{logit}(1 - \delta_{err})) = 1 - \delta_{err}$$

Thus, $1 - \hat{\rho}(\alpha_c) \leq \delta_{err}$. Since $\hat{\rho}(\alpha_c) \leq 1$, this implies $|\hat{\rho}(\alpha_c) - 1| \leq \delta_{err}$. This occurs with probability $\geq 1 - \delta_{\text{sep}}$ for $\alpha_c \sim \mathcal{D}_1$.

**Case 2:** Concept is absent ($\alpha_c \sim \mathcal{D}_0$) and $\alpha_c \leq \beta - \epsilon$. The input to the sigmoid is $X_{\text{abs}} = (\alpha_c - \beta)/\gamma$. Since $\alpha_c - \beta \leq -\epsilon$, we have $X_{\text{abs}} \leq -\epsilon/\gamma$. Substituting the chosen $\gamma$:

$$X_{\text{abs}} \leq \frac{-\epsilon}{\epsilon/\text{logit}(1 - \delta_{err})} = -\text{logit}(1 - \delta_{err}) = \text{logit}(\delta_{err})$$

As $\sigma(\cdot)$ is monotonically increasing:

$$\hat{\rho}(\alpha_c) = \sigma(X_{\text{abs}}) \leq \sigma(\text{logit}(\delta_{err})) = \delta_{err}$$

Thus, $|\hat{\rho}(\alpha_c) - 0| \leq \delta_{err}$. This occurs with probability $\geq 1 - \delta_{\text{sep}}$ for $\alpha_c \sim \mathcal{D}_0$.

This proves that for inputs falling into the high-confidence regions (i.e., outside $(\beta - \epsilon, \beta + \epsilon)$), the estimator $\hat{\rho}(\alpha_c)$ is $\delta_{err}$-close to the ideal binary value. Assumption 3.1 ensures these high-confidence regions are indeed where most samples from $\mathcal{D}_1$ and $\mathcal{D}_0$ lie. $\qquad\square$

*Remark on the uncertainty region $(\beta - \epsilon, \beta + \epsilon)$:* The theorem guarantees performance for $\alpha_c$ values outside this uncertainty margin. Inside this margin, the value of $\hat{\rho}(\alpha_c)$ will smoothly transition between approximately $\delta_{err}$ and $1 - \delta_{err}$. Assumption 3.1 states that the probability of an $\alpha_c$ from either class $\mathcal{D}_0$ or $\mathcal{D}_1$ falling into the *other* class's side of this margin (i.e., $\alpha \sim \mathcal{D}_0$ having $\alpha > \beta - \epsilon$, or $\alpha \sim \mathcal{D}_1$ having $\alpha < \beta + \epsilon$) is at most $\delta_{\text{sep}}$. If $\delta_{\text{sep}}$ is small, misclassifications or uncertain estimations are infrequent. The practical choice of $\beta$ aims to minimize errors within this region based on the observed distributions.

## F  Theoretical Analysis of Latent Concept Persistence (LCP) Mitigation

This appendix provides a formal theoretical framework for the LCP mitigation strategy described in Section 3.3, based on standard assumptions from optimization theory.

### F.1  Formal LCP Risk Definition and Assumptions

Latent Concept Persistence (LCP) occurs when concepts $\mathcal{C}_{\text{erase}}$, intended for removal and presumptively absent from the direct semantics of an initial surgically modified embedding $\hat{e}'_s$ (output of Eq. (16)), are nonetheless generated. This is attributed to the U-Net's visual priors being triggered by other, semantically distinct concepts $c_{\text{im}}$ present in or implied by $\hat{e}'_s$. Let $\mathcal{C}_{\text{target}} \subseteq \mathcal{C}_{\text{erase}}$ be the subset of concepts visually detected as persistent from images generated by $\hat{e}'_s$. We define the LCP risk function $\mathcal{R} : \mathbb{R}^k \to [0, 1]$ for an embedding $e$ (which is assumed to be semantically sanitized w.r.t. $\mathcal{C}_{\text{target}}$ but may still cause their visual generation) as:

$$\mathcal{R}(e) = \mathbb{E}_{I \sim p_\theta(I|e)} \left[ \max_{c \in \mathcal{C}_{\text{target}}} \mathbb{I}(c \in \text{Concepts}(I)) \right]. \tag{21}$$

This measures the probability of at least one concept from $\mathcal{C}_{\text{target}}$ appearing. For analysis, we assume $\mathcal{R}(e)$ or a suitable smooth surrogate is differentiable.

The final LCP-mitigated surgery (Eq. (18)) is $\hat{e}'_{\text{final}} = e_{\text{input}} - \rho_{\text{eff}} \Delta e_{\text{surg}}$, where we map notation:

- $e_0 = e_{\text{input}}$ (the embedding to which the full surgery is applied).
- $e_1 = \hat{e}'_{\text{final}}$ (the embedding after the LCP-aware surgery).
- $\rho_{\text{eff}} = \hat{\rho}^*_{\text{joint}}$ (the effective removal strength from Eq. (17)).
- $\Delta e_{\text{surg}} = \Delta e^*_{\text{co}}$ (the joint removal direction for $\mathcal{C}^* = \mathcal{C}_{\text{sem}} \cup \mathcal{C}_{\text{target}}$).

The risk we aim to reduce is that observed at $\mathcal{R}(\hat{e}'_s)$.

We make the following standard assumptions:

(A1) **Directional Alignment:** The chosen surgery direction $\Delta e_{\text{surg}}$ is well-aligned with the negative gradient of the risk function $\mathcal{R}$ evaluated at $e_0 = e_{\text{input}}$. That is, there exists a constant $\eta_{\mathcal{R}} > 0$ such that:

$$\langle \nabla \mathcal{R}(e_0), \Delta e_{\text{surg}} \rangle \geq \eta_{\mathcal{R}} \|\nabla \mathcal{R}(e_0)\| \|\Delta e_{\text{surg}}\| \tag{22}$$

This implies that subtracting $\rho_{\text{eff}} \Delta e_{\text{surg}}$ from $e_0$ moves the embedding in a direction that tends to reduce $\mathcal{R}$. The construction of $\Delta e_{\text{surg}}$ based on $\mathcal{C}_{\text{target}}$ (visually persistent concepts) strongly supports this alignment for mitigating LCP.

(A2) **L-Smoothness:** The gradient $\nabla \mathcal{R}(e)$ is Lipschitz continuous with constant $L \geq 0$:
$$\|\nabla \mathcal{R}(e_a) - \nabla \mathcal{R}(e_b)\| \leq L\|e_a - e_b\| \quad \forall e_a, e_b. \tag{23}$$

## F.2 LCP Risk Reduction Theorem

**Theorem 3** (LCP Risk Bound via Directional Surgery). *Let $e_0 = e_{input}$ and $e_1 = \hat{e}'_{final} = e_0 - \rho_{eff}\Delta e_{surg}$. Under Assumptions (A1) and (A2), the post-surgery risk $\mathcal{R}(e_1)$ is bounded by:*
$$\mathcal{R}(e_1) \leq \mathcal{R}(e_0) - \rho_{eff}\eta_\mathcal{R}\|\nabla \mathcal{R}(e_0)\|\|\Delta e_{surg}\| + \frac{L(\rho_{eff})^2}{2}\|\Delta e_{surg}\|^2. \tag{24}$$

*For risk reduction (i.e., $\mathcal{R}(e_1) < \mathcal{R}(e_0)$), the effective step size $\rho_{eff}$ must satisfy $\rho_{eff} < \frac{2\eta_\mathcal{R}\|\nabla \mathcal{R}(e_0)\|}{L\|\Delta e_{surg}\|}$.*

*Proof.* This result is a standard consequence of L-smoothness in optimization. For an L-smooth function $\mathcal{R}$, and an update $e_1 = e_0 + \delta e$, Taylor's theorem with a remainder bound (or descent lemma) gives:
$$\mathcal{R}(e_1) \leq \mathcal{R}(e_0) + \langle \nabla \mathcal{R}(e_0), \delta e \rangle + \frac{L}{2}\|\delta e\|^2. \tag{25}$$
In our LCP mitigation surgery, the update is $\delta e = e_1 - e_0 = -\rho_{\text{eff}}\Delta e_{\text{surg}}$. Substituting this into Eq. (25):
$$\mathcal{R}(e_1) \leq \mathcal{R}(e_0) + \langle \nabla \mathcal{R}(e_0), -\rho_{\text{eff}}\Delta e_{\text{surg}} \rangle + \frac{L}{2}\| -\rho_{\text{eff}}\Delta e_{\text{surg}}\|^2$$
$$= \mathcal{R}(e_0) - \rho_{\text{eff}}\langle \nabla \mathcal{R}(e_0), \Delta e_{\text{surg}} \rangle + \frac{L(\rho_{\text{eff}})^2}{2}\|\Delta e_{\text{surg}}\|^2.$$
Now, we apply the Directional Alignment condition (Assumption (A1)): $\langle \nabla \mathcal{R}(e_0), \Delta e_{\text{surg}} \rangle \geq \eta_\mathcal{R}\|\nabla \mathcal{R}(e_0)\|\|\Delta e_{\text{surg}}\|$. Since this term is preceded by a negative sign in Eq. (**??**), we have: $-\rho_{\text{eff}}\langle \nabla \mathcal{R}(e_0), \Delta e_{\text{surg}} \rangle \leq -\rho_{\text{eff}}\eta_\mathcal{R}\|\nabla \mathcal{R}(e_0)\|\|\Delta e_{\text{surg}}\|$. Substituting this back gives:
$$\mathcal{R}(e_1) \leq \mathcal{R}(e_0) - \rho_{\text{eff}}\eta_\mathcal{R}\|\nabla \mathcal{R}(e_0)\|\|\Delta e_{\text{surg}}\| + \frac{L(\rho_{\text{eff}})^2}{2}\|\Delta e_{\text{surg}}\|^2.$$
This completes the proof of Eq. (24). For $\mathcal{R}(e_1)$ to be less than $\mathcal{R}(e_0)$, we require the sum of the second and third terms on the right-hand side to be negative:
$$-\rho_{\text{eff}}\eta_\mathcal{R}\|\nabla \mathcal{R}(e_0)\|\|\Delta e_{\text{surg}}\| + \frac{L(\rho_{\text{eff}})^2}{2}\|\Delta e_{\text{surg}}\|^2 < 0$$
Since $\rho_{\text{eff}} > 0$ and assuming $\|\Delta e_{\text{surg}}\| > 0$, we can divide by $\rho_{\text{eff}}\|\Delta e_{\text{surg}}\|$:
$$-\eta_\mathcal{R}\|\nabla \mathcal{R}(e_0)\| + \frac{L\rho_{\text{eff}}}{2}\|\Delta e_{\text{surg}}\| < 0$$
$$\implies \rho_{\text{eff}} < \frac{2\eta_\mathcal{R}\|\nabla \mathcal{R}(e_0)\|}{L\|\Delta e_{\text{surg}}\|}$$
This provides the condition on $\rho_{\text{eff}}$ for guaranteed risk reduction relative to $\mathcal{R}(e_0)$. $\square$

*Interpretation for LCP Mitigation Effect on $\mathcal{R}(\hat{e}'_s)$:* The theorem formally bounds $\mathcal{R}(\hat{e}'_{final})$ relative to $\mathcal{R}(e_{input})$. The practical goal is to reduce the LCP risk $\mathcal{R}(\hat{e}'_s)$ that was observed after the initial semantic-only surgery. The final surgery (Eq. (18)) is specifically designed based on $\mathcal{C}_{target}$ (visually persistent concepts from $\hat{e}'_s$). Thus, $\Delta e_{\text{surg}}$ and $\rho_{\text{eff}}$ are chosen to counteract the factors leading to $\mathcal{R}(\hat{e}'_s) > 0$. While the direct proof applies to the change from $e_{input}$, the informed construction of the surgery ensures that $\hat{e}'_{final}$ is moved to a region of lower LCP risk compared to the state $\hat{e}'_s$. The simplified statement in the main text, $\mathcal{R}(\hat{e}'_{final}) \leq \mathcal{R}(\hat{e}'_s) - C_1\rho^* + \mathcal{O}((\rho^*)^2)$, heuristically captures this intended consequence.

## F.3 Connection to Method Implementation and Empirical Validation

The directional alignment (A1) is supported by constructing $\Delta e_{\text{surg}}$ (our $\Delta e^*_{\text{co}}$) to explicitly include visually persistent concepts $\mathcal{C}_{\text{target}}$. L-smoothness (A2) is a common modeling assumption for deep learning systems. The step size $\rho_{\text{eff}}$ (our $\hat{\rho}^*_{\text{joint}}$) is effectively controlled by the empirical tuning of $\lambda_{\text{vis}}$ and the estimated concept presence scores. Qualitative examples demonstrating successful LCP removal provide visual confirmation of the method's efficacy.

The example in Figure 7 clearly demonstrates that even when a concept (like "tree") is not semantically cued by the prompt ("a road") and might be missed by semantic-only analysis, LCP can cause its generation due to strong model priors. Our visual feedback mechanism effectively identifies such instances, and the subsequent LCP-aware surgery successfully removes the unintended concept, aligning with the theoretical prediction of risk reduction. This highlights the importance of the visual feedback loop in achieving comprehensive concept erasure.

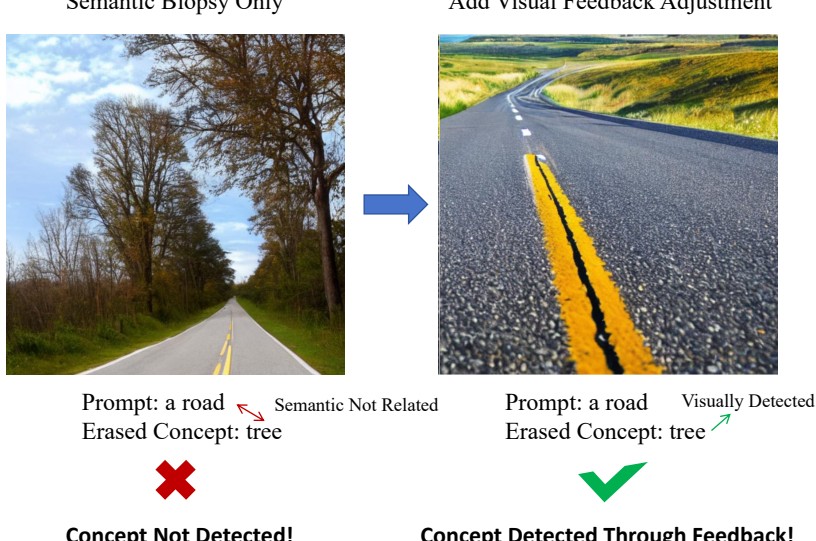

Semantic Biopsy Only      Add Visual Feedback Adjustment

Prompt: a road ⤹ Semantic Not Related
Erased Concept: tree

Prompt: a road   Visually Detected
Erased Concept: tree ↗

❌        ✅

**Concept Not Detected!**      **Concept Detected Through Feedback!**

Figure 7: Qualitative illustration of LCP mitigation for the target concept "tree". (**Left - Semantic Biopsy Only**): The input prompt is "a road", and the "tree" concept is targeted for erasure. Despite the prompt not semantically implying "tree", the initial semantic surgery ($\hat{e}'_s$) fails to prevent trees from appearing, due to the U-Net's strong prior associating roads with trees. Semantic biopsy alone does not detect this LCP issue as the prompt itself is "clean". (**Right - Add Visual Feedback Adjustment**): After generating the image on the left, visual feedback detects the persistent "tree" concept. Our LCP-aware refined surgery (resulting in $\hat{e}'_{\text{final}}$) then incorporates this visual information. The subsequent image generation successfully removes the trees, yielding a scene consistent with only "a road". This visual improvement directly corroborates the risk reduction predicted by Theorem 3.

# G    Theoretical Guarantees for Semantic Surgery

This appendix provides the formal theoretical framework supporting Semantic Surgery, linking our method to the desiderata of Completeness, Locality, and Robustness defined in the main paper (Sec. 3).

## G.1    Core Theoretical Framework and Assumptions

Our analysis relies on three foundational principles, which are empirically validated in our work.

**Assumption G.1** (Statistical Separability)**.** *For any concept $c$, the distribution of cosine similarities $\alpha_c = \cos(\phi(p), \Delta e_c)$ for prompts containing the concept ($\mathcal{D}_1$) is statistically separable from the distribution for prompts not containing the concept ($\mathcal{D}_0$). This is formally stated in the main paper's Assumption 3.1 and empirically validated in Appendix E.2. This property enables high-accuracy concept detection via our sigmoid calibration (Theorem 2), ensuring $|\hat{\rho} - \rho_{ideal}| \leq \delta_{err}$ with high probability.*

**Assumption G.2** (Bounded Linear Erasure Error)**.** *Vector subtraction provides a bounded approximation of ideal semantic removal. For an embedding $e$, its ideal counterpart without concept $c$, $\phi(p \setminus c)$, and the true presence $\rho$, there exists a small, empirically bounded error $\xi$:*
$$\|(e - \rho\Delta e_c) - \phi(p \setminus c)\| \leq \xi.$$
*This is grounded in the linear structure of CLIP's embedding space [30] and is validated by the high efficacy ($Acc_E$) in our experiments.*

**Assumption G.3** (Lipschitz Continuity of the Generative Process)**.** *Small perturbations in the embedding space lead to bounded changes in the probability of generating a concept. Let $g_c(e) = \mathbb{E}_{I \sim p_\theta(I|e)}[\mathbb{I}(c \in Concepts(I))]$. We assume $g_c(e)$ is locally Lipschitz continuous with constant $\lambda_c$:*
$$|g_c(e_1) - g_c(e_2)| \leq \lambda_c \|e_1 - e_2\|.$$

## G.2 Proof of Theoretical Guarantees

### G.2.1 Completeness (Erasure Guarantee)

**Theorem 4** (Completeness). *Let $e' = e_{input} - \hat{\rho}\Delta e_{erase}$ be the modified embedding for a concept $c \in \mathcal{C}_{erase}$. Under Assumptions G.1-G.3, for a chosen safety threshold $\epsilon_{safe}$, if parameters are set such that $\lambda_c(\|\Delta e_{erase}\|\delta_{err} + \xi) < \epsilon_{safe}$, then:*

$$g_c(e') \leq \epsilon_{safe} + O(\delta_{\text{sep}}).$$

*Proof.* Let $e_{\text{ideal}} = \phi(p \setminus c)$ be the ideal embedding without concept $c$. The error between our modified embedding $e'$ and the ideal one is $\|e' - e_{\text{ideal}}\| = \|(e_{input} - \hat{\rho}\Delta e_{erase}) - e_{\text{ideal}}\|$. By applying the triangle inequality and adding/subtracting $\rho\Delta e_{\text{erase}}$ (where $\rho$ is the true presence):

$$\|e' - e_{\text{ideal}}\| = \|(e_{input} - \rho\Delta e_{\text{erase}} - e_{\text{ideal}}) + (\rho - \hat{\rho})\Delta e_{\text{erase}}\|$$
$$\leq \|(e_{input} - \rho\Delta e_{\text{erase}}) - e_{\text{ideal}}\| + \|(\rho - \hat{\rho})\Delta e_{\text{erase}}\|$$
$$\leq \xi + |\rho - \hat{\rho}| \cdot \|\Delta e_{\text{erase}}\| \quad \text{(by Assumption G.2).}$$

From Assumption G.1 and Theorem 2 from the main paper, we know that $|\rho - \hat{\rho}| \leq \delta_{\text{err}}$ with probability $\geq 1 - \delta_{\text{sep}}$. Thus, $\|e' - e_{\text{ideal}}\| \leq \xi + \delta_{\text{err}}\|\Delta e_{\text{erase}}\|$. By Assumption G.3 (Lipschitz continuity of $g_c$):

$$|g_c(e') - g_c(e_{\text{ideal}})| \leq \lambda_c\|e' - e_{\text{ideal}}\| \leq \lambda_c(\xi + \delta_{\text{err}}\|\Delta e_{\text{erase}}\|).$$

By construction, the ideal embedding $e_{\text{ideal}}$ does not contain concept $c$, so $g_c(e_{\text{ideal}}) = 0$. This gives:

$$g_c(e') \leq \lambda_c(\xi + \delta_{\text{err}}\|\Delta e_{\text{erase}}\|).$$

If we choose parameters such that the right-hand side is less than $\epsilon_{\text{safe}}$, the bound holds. The term $O(\delta_{\text{sep}})$ accounts for the small probability that our presence estimation $\hat{\rho}$ falls outside the $\delta_{\text{err}}$ error bound. The LCP mitigation module (main paper, Sec. 3.3) further reduces any residual risk empirically. $\qquad\square$

### G.2.2 Locality (Fidelity Guarantee)

**Theorem 5** (Locality). *For any non-target concept $d \notin \mathcal{C}_{erase}$, the change in its generation probability is bounded. With probability at least $1 - \delta_{sep}$:*

$$|g_d(e') - g_d(e)| = 0,$$

*and with probability at most $\delta_{sep}$, the impact is bounded by a negligible value $\kappa$:*

$$|g_d(e') - g_d(e)| \leq \kappa, \quad \text{where} \quad \kappa = \lambda_d M_{co}.$$

*Proof.* Let $\delta e = e' - e = -\hat{\rho}_{\text{joint}}\Delta e_{\text{co}}$ be the surgery vector applied to the input embedding $e$. The proof proceeds by considering two cases based on whether the surgery is activated.

**Case 1: Surgery is not activated** ($\delta e = 0$). This occurs when no target concepts are detected in the input prompt, i.e., $\hat{\rho}_{\text{joint}} < \tau$ (where $\tau$ is the activation threshold). From Assumption G.1 and Theorem 2, the probability of a false positive detection for any single concept is low. Thus, the probability of incorrectly activating the surgery when no target concepts are present is bounded by $\delta_{\text{sep}}$. Consequently, with a high probability of at least $1 - \delta_{\text{sep}}$, no surgery is applied. In this case, $e' = e$, which means $\delta e = 0$, and the impact on any non-target concept $d$ is zero:

$$|g_d(e') - g_d(e)| = 0.$$

**Case 2: Surgery is activated** ($\delta e \neq 0$). This occurs with a low probability of at most $\delta_{\text{sep}}$ when no target concept is present, or when a target concept is correctly detected. By Assumption G.3 (Lipschitz continuity), the change in the generation probability of concept $d$ is bounded by:

$$|g_d(e') - g_d(e)| \leq \lambda_d\|e' - e\| = \lambda_d\|\delta e\|.$$

We can bound the norm of the surgery vector. Since $\hat{\rho}_{\text{joint}} \in [0, 1]$ and letting $M_{\text{co}} = \|\Delta e_{\text{co}}\|$ be the norm of the co-occurrence direction vector (which is bounded for a given set of concepts), we have:

$$\|\delta e\| = \|\hat{\rho}_{\text{joint}}\Delta e_{\text{co}}\| \leq M_{\text{co}}.$$

Therefore, the maximum impact in this case is bounded by $\kappa = \lambda_d M_{\text{co}}$.

While this provides a formal bound, the practical impact is typically much smaller. This is because the surgery vector $\Delta e_{\text{co}}$ is constructed from target concepts and is thus nearly orthogonal to the semantic direction $\Delta e_d$ of a distinct, unrelated concept $d$. This near-orthogonality ensures that the vector subtraction primarily affects semantic components related to $\mathcal{C}_{\text{erase}}$, minimizing collateral impact on $d$.

**Conclusion.** Combining both cases, the expected impact on a non-target concept is negligible. The most common scenario (with probability $\geq 1 - \delta_{\text{sep}}$ for concept-absent prompts) results in zero impact. In the less frequent case of an incorrect activation, the impact is bounded by a small constant $\kappa$. This proves the locality of our method. $\qquad\square$

### G.2.3 Robustness (Resistance to Paraphrasing)

**Theorem 6** (Robustness as Lipschitz Continuity). *The Semantic Surgery operator $\mathcal{T}(e) = e - \hat{\rho}(e)\Delta e_{co}(e)$ is locally Lipschitz continuous. That is, there exists a constant $L > 0$ such that for any embedding $e$ and a sufficiently small perturbation $\delta e$:*

$$\|\mathcal{T}(e + \delta e) - \mathcal{T}(e)\| \leq L\|\delta e\|.$$

*Proof.* Let $e_1 = e$ and $e_2 = e + \delta e$. We need to bound $\|\mathcal{T}(e_2) - \mathcal{T}(e_1)\|$.

$$\|\mathcal{T}(e_2) - \mathcal{T}(e_1)\| = \|(e_2 - \hat{\rho}(e_2)\Delta e_{co}(e_2)) - (e_1 - \hat{\rho}(e_1)\Delta e_{co}(e_1))\|$$
$$= \|(e_2 - e_1) - (\hat{\rho}(e_2)\Delta e_{co}(e_2) - \hat{\rho}(e_1)\Delta e_{co}(e_1))\|$$
$$\leq \|e_2 - e_1\| + \|\hat{\rho}(e_2)\Delta e_{co}(e_2) - \hat{\rho}(e_1)\Delta e_{co}(e_1)\|.$$

The first term is $\|\delta e\|$. We focus on the second term by adding and subtracting $\hat{\rho}(e_1)\Delta e_{co}(e_2)$:

$$\|\hat{\rho}(e_2)\Delta e_{co}(e_2) - \hat{\rho}(e_1)\Delta e_{co}(e_1)\| \leq |\hat{\rho}(e_2) - \hat{\rho}(e_1)|\|\Delta e_{co}(e_2)\| + |\hat{\rho}(e_1)|\|\Delta e_{co}(e_2) - \Delta e_{co}(e_1)\|.$$

Let's bound each component:

1. $\hat{\rho}(e) = \sigma((\cos(e, \Delta e_c) - \beta)/\gamma)$ is a composition of locally Lipschitz functions. The cosine similarity $\cos(e, v)$ is locally Lipschitz w.r.t. $e$, and the sigmoid $\sigma$ is globally Lipschitz. Thus, $\hat{\rho}(e)$ is locally Lipschitz with some constant $L_\rho$, so $|\hat{\rho}(e_2) - \hat{\rho}(e_1)| \leq L_\rho\|e_2 - e_1\| = L_\rho\|\delta e\|$.
2. The concept direction norm $\|\Delta e_{co}(e_2)\|$ is bounded by some constant $M$.
3. The co-occurrence direction $\Delta e_{co}(e)$ is also locally Lipschitz. This is because it depends on which concepts exceed the threshold $\tau$, but within a neighborhood where the active set of concepts does not change, $\Delta e_{co}$ is constant. At the boundaries, it has jump discontinuities, but we consider a neighborhood where it is stable. For simplicity, assume it is locally Lipschitz with constant $L_{co}$, so $\|\Delta e_{co}(e_2) - \Delta e_{co}(e_1)\| \leq L_{co}\|\delta e\|$.
4. The term $|\hat{\rho}(e_1)|$ is bounded by 1.

Substituting these bounds back:

$$\|\mathcal{T}(e_2) - \mathcal{T}(e_1)\| \leq \|\delta e\| + (L_\rho\|\delta e\|)M + (1)(L_{co}\|\delta e\|) = (1 + ML_\rho + L_{co})\|\delta e\|.$$

Letting $L = 1 + ML_\rho + L_{co}$, we have shown that $\mathcal{T}$ is locally Lipschitz continuous, which proves the theorem. This directly guarantees that small changes in input embeddings lead to proportionally small changes in the output, ensuring robustness. $\square$

## H    Additional Evaluation Results on CIFAR10 Classes

This section provides the complete evaluation results for object erasure across all 10 CIFAR-10 classes, supplementing the selected results presented in the main paper (Table 1). The metrics $Acc_E$ (Efficacy), $Acc_R$ (Robustness), $Acc_L$ (Locality), and $H$ (Harmonic Mean) are defined in Section 4. As shown in Table 9, Semantic Surgery consistently demonstrates strong performance across all classes, achieving the highest average $H$ score.

## I    Detailed Analysis of Adversarial Robustness

To address the critical concern of adversarial robustness, we evaluated Semantic Surgery against both black-box and white-box adversarial prompt attacks, which are designed to circumvent erasure mechanisms and regenerate forbidden concepts [50, 59].

**Experimental Setup.**    We conducted two sets of experiments. **(1) Black-Box Attack:** We used the Ring-A-Bell (RAB) benchmark [50], a model-agnostic attack that generates adversarial prompts through linguistic manipulation. To ensure statistical significance, we expanded the test set to 380 adversarial prompts targeting object erasure. We compared our method against both training-free (SLD, SAFREE) and parameter-modifying (MACE, Receler) baselines. **(2) White-Box Attack:** We adapted the UnlearnDiffAtk framework [59], an optimization-based attack that directly manipulates embeddings in a model-aware manner, to test our method's resilience against a gradient-based search for vulnerabilities. For both setups, we measure the Attack Success Rate (ASR), where lower is better. Our core method (Semantic Surgery without LCP) was used for these tests.

**Detailed Analysis.**    As shown in Table 4 in the main paper, Semantic Surgery exhibits state-of-the-art adversarial robustness. Against the large-scale black-box RAB attack, our method achieves an ASR of just **1.05%**, a 3.7x reduction compared to the strongest baseline, MACE (3.95%). A one-sided Fisher's Exact Test on this 380-sample set confirms this difference is highly statistically significant (p=0.0089).

Table 9: Full evaluation of object erasure on all CIFAR-10 classes. $Acc_E$: Efficacy (lower is better), $Acc_R$: Robustness (lower is better), $Acc_L$: Locality (higher is better), H: Harmonic Mean (higher is better).

| Classes | Metric | SD1.4 | ESD-x | ESD-u | AC | UCE | Receler | MACE | Ours |
|---------|--------|-------|-------|-------|-----|-----|---------|------|------|
| **airplane** | $Acc_E\downarrow$ | 100.00 | 30.00 | 12.00 | 2.00 | 10.00 | 4.00 | 0.00 | 2.00 |
| | $Acc_R\downarrow$ | 70.00 | 62.00 | 24.00 | 18.00 | 34.00 | 6.00 | 10.00 | 4.00 |
| | $Acc_L\uparrow$ | 89.11 | 87.89 | 86.44 | 87.44 | 90.78 | 87.33 | 85.56 | 89.11 |
| | H↑ | - | 57.72 | 83.13 | 88.67 | 80.48 | 92.29 | 91.47 | 94.21 |
| **automobile** | $Acc_E\downarrow$ | 96.00 | 30.00 | 24.00 | 0.00 | 0.00 | 3.00 | 0.00 | 0.00 |
| | $Acc_R\downarrow$ | 84.00 | 74.00 | 64.00 | 24.00 | 54.00 | 18.00 | 18.00 | 4.00 |
| | $Acc_L\uparrow$ | 87.56 | 87.44 | 88.22 | 83.78 | 85.11 | 84.00 | 83.11 | 79.78 |
| | H↑ | - | 46.74 | 57.39 | 85.48 | 68.98 | 87.19 | 87.65 | 91.04 |
| **bird** | $Acc_E\downarrow$ | 87.56 | 11.00 | 10.00 | 0.00 | 4.00 | 1.00 | 0.00 | 0.00 |
| | $Acc_R\downarrow$ | 100.00 | 84.00 | 50.00 | 82.00 | 62.00 | 26.00 | 24.00 | 2.00 |
| | $Acc_L\uparrow$ | 90.00 | 84.56 | 78.00 | 87.44 | 85.67 | 80.22 | 74.89 | 86.89 |
| | H↑ | - | 35.06 | 68.29 | 38.97 | 61.98 | 83.15 | 82.17 | 94.60 |
| **cat** | $Acc_E\downarrow$ | 97.00 | 18.00 | 4.00 | 0.00 | 1.00 | 0.00 | 0.00 | 1.00 |
| | $Acc_R\downarrow$ | 98.00 | 46.00 | 26.00 | 42.00 | 6.00 | 0.00 | 18.00 | 0.00 |
| | $Acc_L\uparrow$ | 86.00 | 84.33 | 78.44 | 86.11 | 83.56 | 80.22 | 83.33 | 86.00 |
| | H↑ | - | 70.47 | 81.79 | 77.21 | 91.72 | 92.41 | 87.73 | 94.55 |
| **deer** | $Acc_E\downarrow$ | 99.00 | 7.00 | 6.00 | 6.00 | 2.00 | 0.00 | 0.00 | 1.00 |
| | $Acc_R\downarrow$ | 96.00 | 60.00 | 32.00 | 68.00 | 2.00 | 0.00 | 0.00 | 0.00 |
| | $Acc_L\uparrow$ | 86.22 | 82.78 | 71.11 | 82.89 | 84.78 | 72.22 | 72.22 | 84.44 |
| | H↑ | - | 62.72 | 76.13 | 55.60 | 93.16 | 88.64 | 88.64 | 93.92 |
| **dog** | $Acc_E\downarrow$ | 99.00 | 32.00 | 14.00 | 0.00 | 1.00 | 0.00 | 0.00 | 2.00 |
| | $Acc_R\downarrow$ | 92.00 | 72.00 | 40.00 | 66.00 | 30.00 | 2.00 | 16.00 | 0.00 |
| | $Acc_L\uparrow$ | 86.67 | 83.22 | 77.56 | 84.89 | 83.67 | 75.78 | 78.89 | 86.44 |
| | H↑ | - | 48.05 | 72.84 | 58.60 | 82.56 | 89.82 | 86.75 | 94.42 |
| **frog** | $Acc_E\downarrow$ | 100.00 | 12.00 | 3.00 | 0.00 | 1.00 | 0.00 | 0.00 | 4.00 |
| | $Acc_R\downarrow$ | 88.00 | 34.00 | 20.00 | 24.00 | 4.00 | 0.00 | 2.00 | 2.00 |
| | $Acc_L\uparrow$ | 87.11 | 85.22 | 80.89 | 85.44 | 87.11 | 82.89 | 70.00 | 86.89 |
| | H↑ | - | 78.43 | 85.30 | 86.05 | 93.76 | 93.56 | 86.98 | 93.37 |
| **horse** | $Acc_E\downarrow$ | 100.00 | 12.00 | 9.00 | 0.00 | 0.00 | 0.00 | 0.00 | 0.00 |
| | $Acc_R\downarrow$ | 96.00 | 62.00 | 48.00 | 44.00 | 18.00 | 4.00 | 6.00 | 0.00 |
| | $Acc_L\uparrow$ | 86.22 | 84.78 | 83.11 | 86.67 | 81.56 | 80.22 | 75.56 | 82.22 |
| | H↑ | - | 60.64 | 71.00 | 76.15 | 87.07 | 91.24 | 88.56 | 93.28 |
| **ship** | $Acc_E\downarrow$ | 100.00 | 40.00 | 30.00 | 25.00 | 4.00 | 17.00 | 4.00 | 3.00 |
| | $Acc_R\downarrow$ | 70.00 | 64.00 | 46.00 | 58.00 | 46.00 | 32.00 | 18.00 | 6.00 |
| | $Acc_L\uparrow$ | 89.11 | 88.56 | 87.78 | 86.33 | 88.56 | 89.11 | 89.11 | 89.11 |
| | H↑ | - | 53.82 | 67.88 | 61.57 | 74.58 | 79.00 | 88.67 | 93.26 |
| **truck** | $Acc_E\downarrow$ | 100.00 | 30.00 | 13.00 | 0.00 | 0.00 | 0.00 | 0.00 | 2.00 |
| | $Acc_R\downarrow$ | 88.00 | 74.00 | 44.00 | 50.00 | 26.00 | 12.00 | 26.00 | 2.00 |
| | $Acc_L\uparrow$ | 87.11 | 86.11 | 87.11 | 84.33 | 84.22 | 83.78 | 78.22 | 84.67 |
| | H↑ | - | 46.61 | 73.47 | 71.67 | 84.78 | 90.09 | 82.65 | 93.11 |
| **Avg** | $Acc_E\downarrow$ | 99.10 | 22.20 | 12.50 | 3.30 | 2.30 | 2.50 | **0.40** | 1.50 |
| | $Acc_R\downarrow$ | 87.20 | 63.20 | 39.40 | 47.60 | 28.20 | 10.00 | 13.80 | **2.00** |
| | $Acc_L\uparrow$ | 87.33 | 85.49 | 81.87 | 85.53 | 85.50 | 81.58 | 79.09 | **85.56** |
| | H↑ | - | 56.03 | 73.72 | 70.00 | 81.90 | 88.74 | 87.13 | **93.58** |

Even more notably, our method achieved a **0.0% ASR** against the white-box UnlearnDiffAtk. We hypothesize this exceptional resilience stems from our method's core "binary gate" mechanism. The Semantic Biopsy step (Sec. 3.2) creates a sharp decision boundary based on cosine similarity ($\alpha_c$). An adversarial attack faces a difficult optimization challenge: it must craft an embedding that contains the target concept visually but remains on the "concept-absent" side of the decision threshold $\beta$ semantically, a contradictory objective that proved insurmountable for the attacks tested.

Interestingly, we observed that while the white-box attack failed to regenerate the concept, it often produced images with significant quality degradation. This reveals a secondary benefit: our framework

acts as a **built-in threat detection system**. An attacker's attempt to find a bypass consistently triggers a high concept presence score ($\hat{\rho}$), even if the final image is corrupted. By monitoring high-$\hat{\rho}$ flags for erased concepts at the pre-generation stage, a system can proactively detect and log adversarial probing attempts without needing to analyze the final visual output.

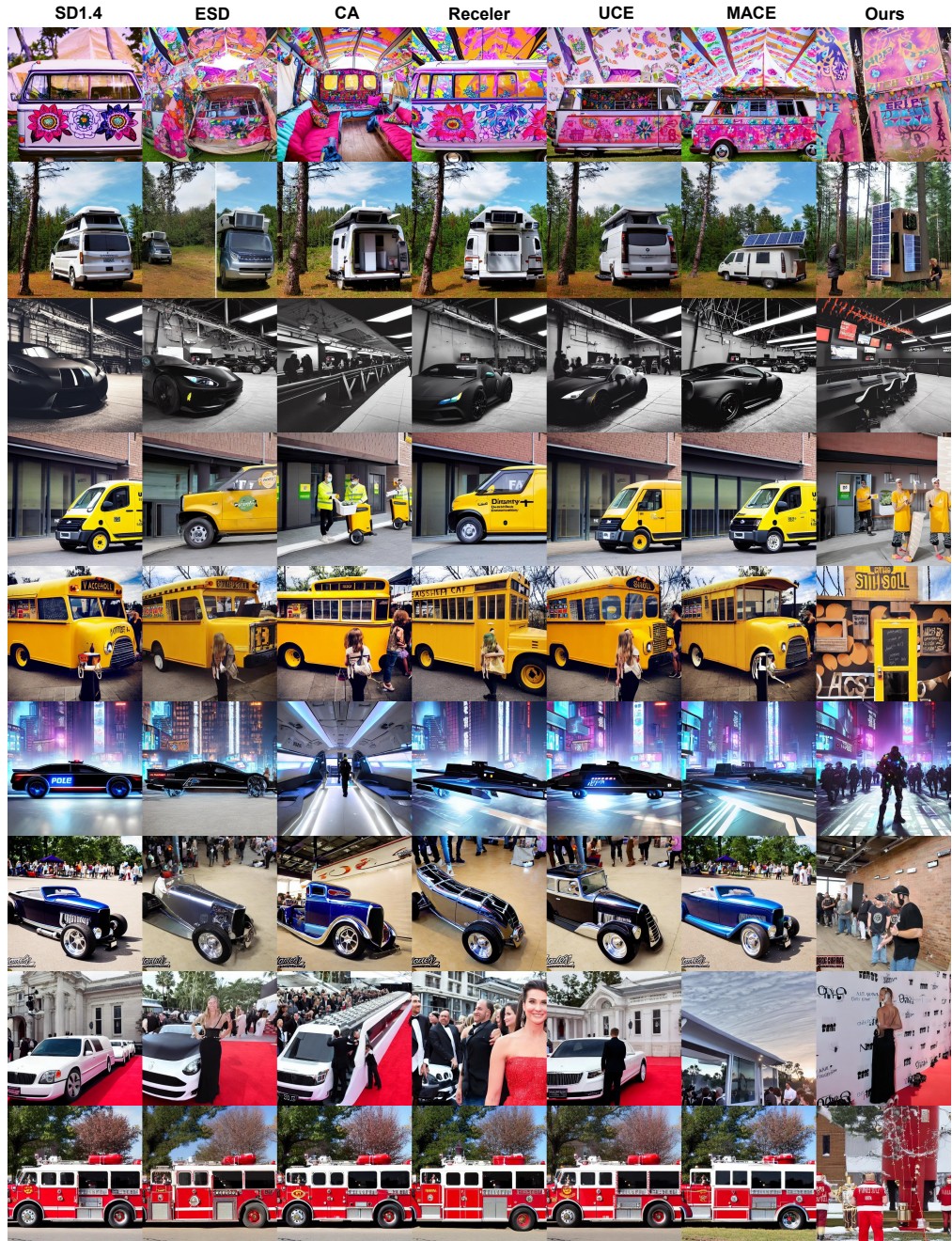

Figure 8: Qualitative comparison of Completeness On CIFAR10.(erased automobile)

## J   Qualitative Results

This section provides qualitative examples illustrating the performance of Semantic Surgery. Figure 9 shows side-by-side comparisons for various erasure tasks including object (automobile, bird, ship), explicit content (Nude, showing a safe alternative), style (Van Gogh). These examples demonstrate the method's ability to effectively remove the target concept while preserving the overall image

structure and non-target elements. Figure 10 and Figure 8 specifically highlights the locality and the completeness of our method on CIFAR-10 object erasure. The samples are randomly chosen from the erasure group and the remaining group. As our method successfully remove the targeted concepts while preserving most of the unrelated semantics remaining in the image, our method succeed in eliminating the targeted concepts with accurate semantic surgery. Figure 11 provides examples of artistic style erasure . The generated images successfully remove the characteristic stylistic features of the target artist while retaining the subject matter described in the prompt. Figure 12 demonstrates celebrity erasure. Images generated for erased celebrities do not depict the individuals, while images for retained celebrities are generated correctly. These qualitative results visually corroborate the quantitative findings presented in the main paper, showcasing the effectiveness and precision of Semantic Surgery.

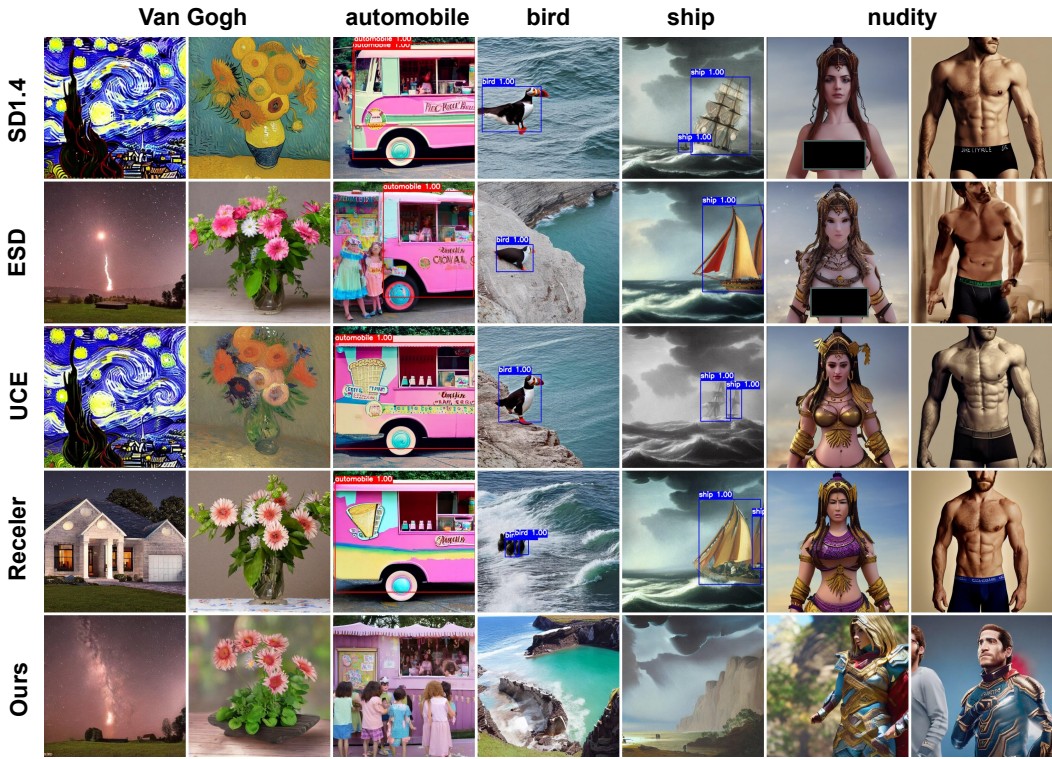

Figure 9: Qualitative comparison across different concept erasure tasks.

## K   Limitations and Future Work

While Semantic Surgery demonstrates strong performance, it has several limitations that open avenues for future work:

- **Dependence on Text Encoder's Linear Structure:** The efficacy of our approach is fundamentally conditioned on the text encoder (e.g., CLIP) exhibiting a near-linear structure where concepts are semantically separable. This assumption, while holding for many common concepts, may be less effective for: (a) future, potentially non-linear encoder architectures, or (b) highly entangled, abstract, or metaphorical concepts (e.g., "a feeling of nostalgia") where a clean vector representation is ill-defined.

- **Sources of Non-Local Effects:** Although our method shows strong locality, it is not entirely free from non-local effects (as seen in the $Acc_L$ metric). We identify three primary sources for this:
    - *Imperfect Semantic Thresholding:* The decision boundary defined by $\beta$ may not be perfectly sharp for every prompt-concept pair. In ambiguous cases, our method might perform a partial semantic operation, inadvertently affecting related but untargeted semantics.
    - *Visual Detector Errors (in LCP):* This is a significant factor. For the optional LCP module, false positives from the vision detector can trigger an unnecessary second-stage erasure,

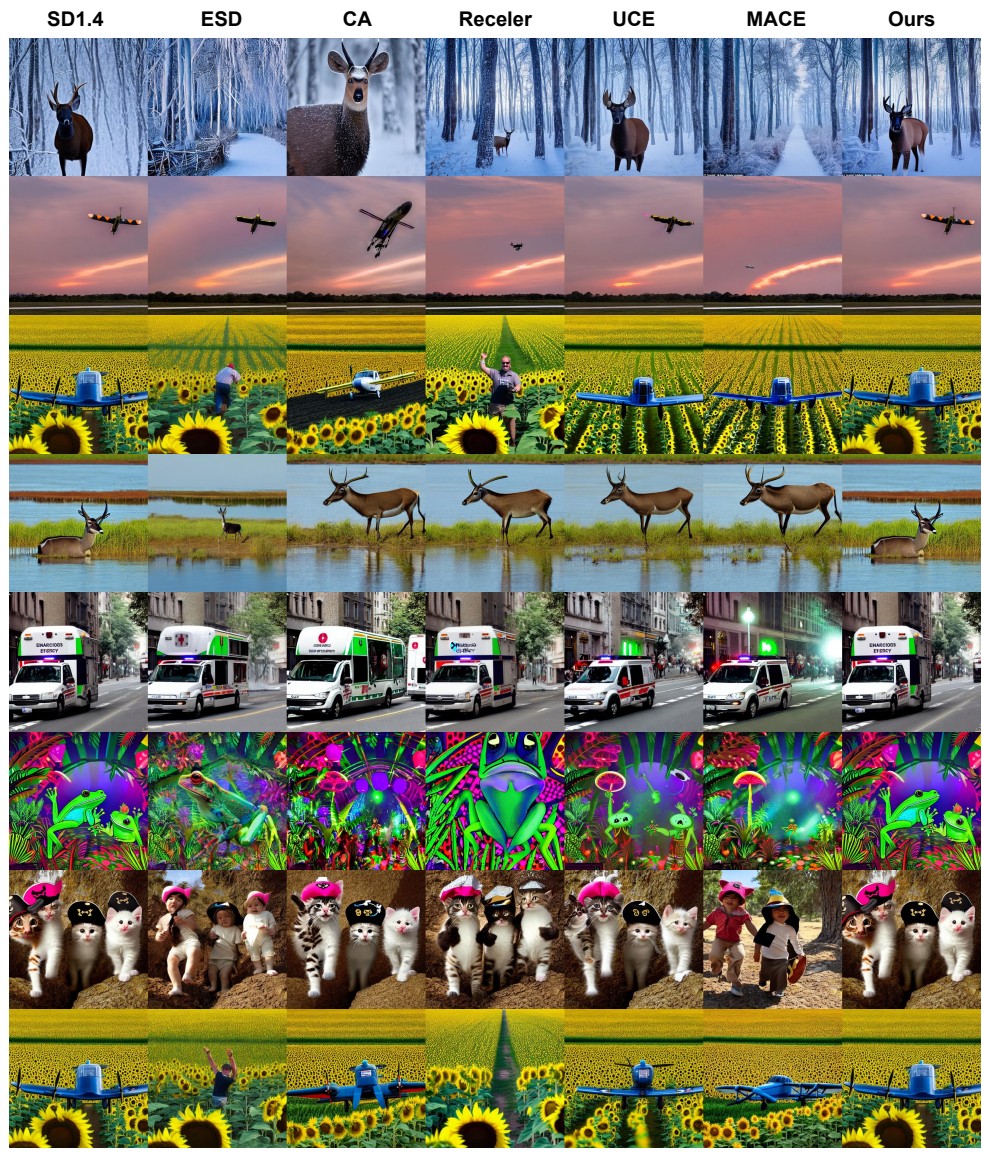

Figure 10: Qualitative comparison of locality on CIFAR-10.

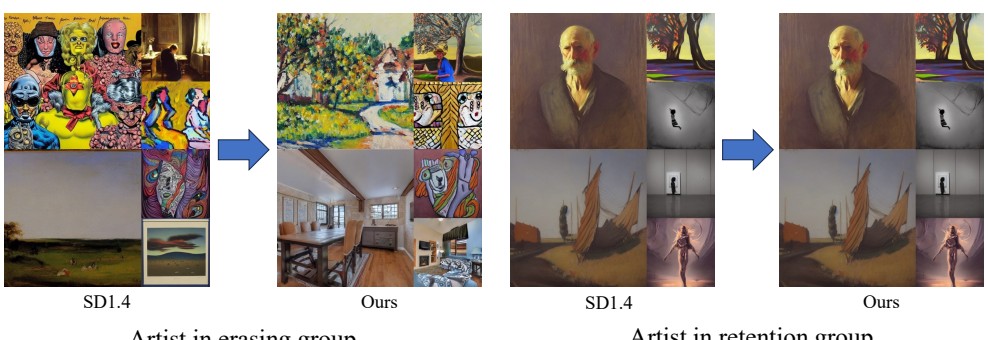

Artist in erasing group

Artist in retention group

Figure 11: Qualitative comparison on Artist Removal.

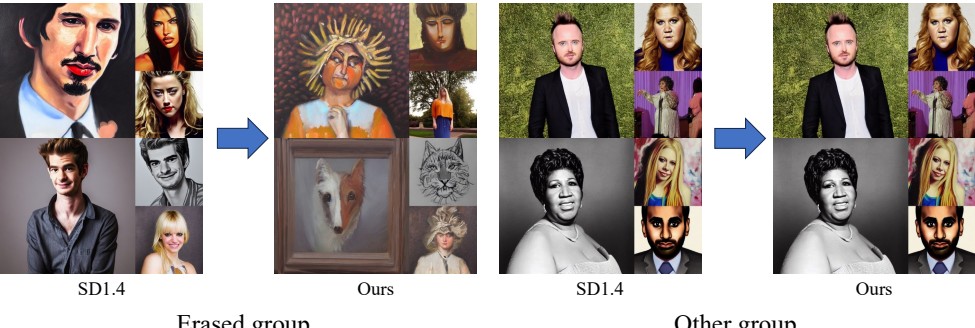

SD1.4         Ours         SD1.4         Ours

Erased group             Other group

Figure 12: Qualitative comparison on Celebrity Removal. The target celebrity is erased from the generated image, while non-target celebrities or general scenes are generated correctly.

> harming locality. For example, when erasing "automobile," the AOD detector sometimes misclassifies "trucks" as "automobiles," causing our LCP module to incorrectly trigger a stronger erasure on prompts that should have generated a "truck." This highlights the critical dependence of the LCP's locality on the detector's precision.
>
> – *Inherent Model Stochasticity:* Some locality degradation is attributable to the inherent stochasticity of the base generative model itself, which can fail to generate concepts perfectly aligned with the prompt even without any intervention.

- **Detector Dependency for LCP Module:** The performance of our optional LCP module is inherently tied to the availability and accuracy of an external visual detector. The lack of reliable detectors for abstract concepts (e.g., complex artistic styles) limits the module's applicability to more concrete, visually verifiable concepts. Furthermore, the detector's performance creates a trade-off: false negatives can lead to incomplete erasure, while false positives can harm locality by triggering incorrect feedback, as discussed above.

- **Scalability to Newer Models:** Our experiments were conducted on Stable Diffusion v1.4 to ensure a fair and direct comparison with the extensive list of prior works in concept erasure [10, 27, 19], which almost exclusively use it as a standard benchmark. Validating the performance and adapting the calibration of Semantic Surgery for newer, more powerful models like SDXL or SD3 is a crucial next step for future work.

Future work could focus on developing adaptive hyperparameter tuning mechanisms, exploring LCP mitigation strategies that are less reliant on external detectors, and investigating methods to further improve concept disentanglement in the embedding space for even more precise multi-concept erasure.

## L    Broader Impact

Semantic Surgery offers a promising approach for enhancing the safety and controllability of text-to-image diffusion models.

**Potential Positive Societal Impacts:**

- **Harmful Content Reduction:** The primary positive impact is the ability to effectively remove undesirable concepts such as explicit content (as demonstrated by near-perfect nudity removal), hate speech related symbols (if properly defined as concepts), or violent imagery. This can make generative AI tools safer for wider public use and reduce the proliferation of harmful synthetic media.

- **Copyright and Intellectual Property Protection:** The method can be used to erase copyrighted artistic styles, characters, or specific celebrity likenesses, helping to mitigate infringement concerns associated with generative models. This could foster more ethical use of AI in creative industries.

- **Bias Mitigation:** If biases (e.g., stereotypical associations) can be identified and represented as semantic concepts, Semantic Surgery could potentially be used to neutralize these biases in

generated images, leading to fairer and more equitable AI systems. For example, removing a concept that a model over-associates with a particular demographic.

- **Enhanced User Control and Customization:** Beyond safety, the ability to precisely remove concepts gives users finer-grained control over image generation, allowing for more creative and specific outputs by excluding unwanted elements.

- **Reduced Need for Costly Retraining:** As a training-free method, it offers a more agile and cost-effective way to update safety protocols or adapt to new content restrictions compared to retraining large diffusion models.

**Potential Negative Societal Impacts and Misuse:**

- **Over-Erasure and Censorship Concerns:** If not carefully calibrated or if applied too broadly, concept erasure techniques could lead to over-erasure, stifling creative expression or unintentionally removing benign content that is semantically close to a target concept. Defining what constitutes an "undesirable" concept can be subjective and culturally dependent, raising concerns about who decides what is erased and the potential for censorship.

- **Adversarial Attacks and Circumvention:** Like any safety mechanism, Semantic Surgery might be susceptible to adversarial attacks designed to bypass its erasure capabilities. Malicious actors could attempt to craft prompts or manipulate embeddings in ways that circumvent the concept detection or neutralization process.

- **False Sense of Security:** While effective, no erasure method is likely to be 100% foolproof against all possible prompts or concept variations. Over-reliance on such tools without acknowledging their limitations could lead to a false sense of security regarding the safety of generative models.

- **Impact on Artistic Expression or Fair Use:** While useful for IP protection, aggressive erasure of artistic styles could also limit transformative uses or artistic exploration that might fall under fair use or be part of legitimate artistic critique or parody.

- **Arms Race in Generative AI Safety:** The development of erasure techniques might contribute to an "arms race" where methods to generate problematic content and methods to block it continually evolve, requiring ongoing research and adaptation.

**Mitigation Strategies:** To mitigate potential negative impacts, several strategies can be considered:

- **Transparency and User Control:** Users should be aware when concept erasure is being applied and, where appropriate, have some control over its application or intensity.

- **Careful Policy and Guideline Development:** The definition of concepts to be erased should be guided by clear, ethical, and transparent policies, ideally developed with community input.

- **Robustness Testing and Red Teaming:** Continuously test the system against adversarial attacks and diverse prompts to identify and address vulnerabilities.

- **Combining with Other Safety Measures:** Semantic Surgery should be seen as one layer in a multi-faceted approach to AI safety, complemented by dataset filtering, model alignment techniques, and post-hoc detection.

- **Research into Explainability:** Further research into why certain concepts are detected or missed can help improve the precision and fairness of the erasure process.

Overall, Semantic Surgery is a tool with significant potential for positive impact, but like all powerful technologies, its deployment requires careful consideration of ethical implications and potential misuse.

