# OpenReview forum: "Semantic Surgery: Zero-Shot Concept Erasure in Diffusion Models"
_NeurIPS.cc/2025/Conference — NeurIPS 2025 poster_

### Official Review · Reviewer_x355 · 2025-06-22

**Clarity:** 2
**Significance:** 2
**Originality:** 3
**Rating:** 4
**Confidence:** 4

**Summary:**

The paper proposes a novel training-free method for removing undesired concepts on the text-embedding level space to improve completeness-locality trade-off in concept erasing for diffusion models. Unlike finetuning methods that retrain or edit the model, the proposed method, Semantic Surgery, operates purely at inference time by manipulating the text embedding before it enters the diffusion pipeline. The key idea is to perform a calibrated vector subtraction that neutralizes the semantic influence of the target concept. To achieve this, Semantic Biopsy estimates how strongly a concept is represented in the prompt compared the text embeddings of target concepts, which is measured as a confidence score through a sigmoid function. For multi-concept erasure, Co-Occurrence Encoding constructs a joint semantic direction for concept removal while avoiding excessive overlap or semantic distortion. To address Latent Concept Persistence (LCP), they introduce a Visual Feedback Loop using detectors like NudeNet or AOD to identify and suppress residual traces of the target concept in generated images. Experiments across four tasks like object erasure, explicit content removal, artistic style erasure, and multi-celebrity erasure demonstrate the state-of-the-art performance on the trade-off between completeness for erasing target concepts and locality for preserving remaining concepts.

**Questions:**

Strengths and Weaknessses include questions

**Ethical Concerns:**

["NO or VERY MINOR ethics concerns only"]

**Final Justification:**

This work introduces a simple yet novel training-fee method for concept erasing using arithmetic operations on text embeddings. The authors provided a new metric called concept presence intensity ($\rho$), along with a rigorous theoretical foundation for its estimation. The method shows clear performance gains on a wide range of concept erasing benchmarks. Although initial submission raised concerns about clarity on presentation, insufficient baselines, lack of experiments on robustness, and ablation studies, the authors addressed most of the concerns by the rebuttal. Therefore, the reviewer raises the score from 3 to 4.

**Limitations:**

yes

**Paper Formatting Concerns:**

No paper formatting concerns

**Quality:**

3

**Strengths And Weaknesses:**

**Strength**
1. The paper proposes a simple yet novel approach for concept erasing based on arithmetic operations on text embeddings. This method is training-free and does not require any computational cost for training to achieve concept erasure.
2. In situations where the presence of target concepts in the prompt is unknown, the authors introduce a novel measure called concept presence intensity $\rho$, which estimates whether the concepts exist. They also provide a rigorous theoretical derivation to support an effective estimation method for this measure.
3. The proposed method demonstrates clear performance improvements on existing extensive concept erasing benchmarks, including objects, explicit content, artistic styles, and celebrities. In particular, the results on I2P benchmarks and multi-concept celebrity erasure tasks exhibit an impressive balance in the completeness-locality trade-off.

** **

**Weaknesses**
1. Without any visual guidance for the explanation of the proposed method, the presentation appears to omit core explanations. For instance, the distinctions between $e _{input}$, $e _{erase}$, $e^{'} _{input}$, $e^{'} _{final}$ need to be described more clearly. In particular, it is stated that the vision detector $\mathcal{D}$ is used to detect the LCP-specific occurrence of target concepts in the generated image. Does this mean that sequential generations of the image from the sample prompts are required? If so, it could significantly increase inference time, which is not well reported in this work. Additionally, the process for construction of $C _{vis}$ should be described more concretely.
2. The method is an inference-time concept erasure but it is never compared with baselines of a similar nature, such as SLD or SAFREE. To validate the strength of the approach as a training-free method, comparisons with these baselines could be included.
3. One crucial concern about this work is the robustness of the proposed method against adversarial attacks. Red-teaming tools such as RAB [1] and UnlearnDiffAttack [2] have shown that several concept erasure approaches in diffusion models are vulnerable to adversarial prompt attacks, allowing erased concepts to be easily regenerated. Recent works on safety in T2I diffusion models—such as Receler, CPE [3], and AdvUnlearn—highlight the importance of addressing this issue. The lack of any such consideration in this work raises concerns regarding the method’s robustness to adversarial scenarios.
4. Ablation studies are lacking: Although the proposed method includes several important components, the paper does not provide corresponding ablation analyses. Key elements that should be ablated include:
    1. The use of Eq. (9) for calibrated presence estimation vs. naive vector addition $\sum _{i=1} ^n \rho _{c _{i}} \Delta e _{c _{i}}$.
    2. The impact of visual feedback for LCP mitigation.
    3. The influence of the threshold parameter $\tau$ on the method’s performance, which should be explored and reported.


I am happy to raise my score if my concerns could be well addressed.

**References**

[1] Tsai et al. "Ring-A-Bell! How Reliable are Concept Removal Methods For Diffusion Models?." *The Twelfth International Conference on Learning Representations*. 2024.

[2] Zhang et al. "To generate or not? safety-driven unlearned diffusion models are still easy to generate unsafe images... for now." *European Conference on Computer Vision*. Cham: Springer Nature Switzerland, 2024.

[3] Lee et al. "Concept pinpoint eraser for text-to-image diffusion models via residual attention gate." *The Thirteenth International Conference on Learning Representations*. 2025.

---

> ### Author Rebuttal · Authors · 2025-07-30
>
> We sincerely thank you for the exceptionally detailed and highly constructive review. Your comments have provided a clear roadmap for improving our paper, and we are especially grateful for your offer to raise your score. We are pleased to report that we have conducted **four new sets of experiments** during the rebuttal period that specifically address every one of your concerns. We are confident that these new results and clarifications will resolve your questions.
>
>
> ### Weakness 1: Clarity, LCP process, inference time, and C_vis construction.
> Thank you for these detailed questions. We apologize for the lack of clarity in our initial submission and have worked to significantly improve the presentation.
>
> **For Visual Explanation:** We have prepared a new diagram that visually explains our entire workflow. This has been included in Section 3 of the final paper to make our method much more intuitive.
>
> **For LCP Process & Detailed Inference Time Analysis:**
> To clarify, the LCP feedback loop is an on-demand, two-pass process. A second, time-consuming pass is **only triggered if** residual concepts are detected. To quantify this overhead, we have **conducted a new, detailed timing experiment**:
>
> | Scenario                           | Time per Image |
> |------------------------------------|----------------|
> | 1. Baseline SDv1.4                 | 3.11 s         |
> | 2. SLD                             | 4.07 s         |
> | 3. SAFREE                          | 3.96 s         |
> | 4. Ours (LCP disabled)             | 3.21 s         |
> | 5. Ours (LCP worst-case, Pass 2)   | 6.43 s         |
> | 6. Ours (LCP avg on I2P)           | **4.09 s**     |
>
> This analysis reveals three key points:
>
> (1) Our core method adds negligible overhead (3.21s vs 3.11s).
>
> (2) The "doubled time" is a worst-case scenario.
>
> (3) In a practical safety task (I2P), our average time (4.09s) is comparable to other training-free methods like SLD and SAFREE, while achieving vastly superior safety (as shown in our new I2P comparison). The LCP module is an optional(only in high demand when security requirements are extremely high), highly efficient trade-off for critical safety needs.
>
> This complete timing analysis has been prepared for inclusion in the final paper.
>
> **For C_vis Construction**: To improve clarity, we have expanded the description in Section 3.3. The revised text now explicitly states that C_vis is formed by selecting concepts {c_i} where the vision detector's confidence score surpasses a threshold τ_vis, and we emphasize that this is an efficient, single-batch operation. This clarification will be present in the final revised version.
>
>
> ### Weakness 2: Missing comparison with baselines like SLD or SAFREE.
> This is a very important point, and we thank you for raising it. Our initial experiments focused on comparing against strong **parameter-modifying** baselines (like MACE and Receler). The rationale was based on the general understanding in the field that inference-time methods, while flexible, typically do not match the performance of methods that directly edit model weights. Our primary goal was therefore attempting to demonstrate that our inference-time approach could not only compete with but **surpass** these powerful, heavyweight baselines, which would be a significant finding. However, to validate your point and provide a complete picture of our method's standing among its direct, training-free peers, we have conducted a **new, head-to-head comparison with SLD and SAFREE on the challenging I2P safety task.**
> The results, broken down by category, are as follows:
>
> **New Comparison on I2P (Unsafe Instances Count ↓):**
> | Method | Armpits | Belly | Buttocks | Feet | Breasts (F) | Genitalia (F) | Breasts (M) | Genitalia (M) | Total |
> |--------|---------|-------|----------|------|-------------|---------------|-------------|---------------|-------|
> | SAFREE | 11      | 22    | 5        | 9    | 15          | 4             | 15          | 1             | 82    |
> | SLD    | 18      | 48    | 7        | 4    | 57          | 15            | 0           | 0             | 149   |
> | Ours   | 0       | 0     | 1        | 0    | 0           | 0             | 0           | 0             | 1     |
>
> This detailed breakdown shows that our method achieves near-perfect erasure across almost all categories, reducing the total number of unsafe generations by **over 98%** compared to the next best training-free method, SAFREE(from 82 instances to 1). This confirms that our method holds a substantial advantage within its category on safety-critical tasks. We have already drafted the analysis for this new comparison and will ensure the table and discussion are included in the final paper.
>
>
> ### Weakness 3: Lack of robustness evaluation against adversarial attacks.
> We agree that adversarial robustness is a critical concern. Our initial work focused on establishing a general-purpose framework for high-fidelity erasure under standard conditions, similar to other general-purpose methods like MACE (CVPR 2024) and AdaVD (CVPR 2025) that didn't add the adversarial experiment as well. However, to address your valid concern, we conducted a **new, targeted experiment** using the **Ring-A-Bell (RAB) benchmark** you referenced. We created a test set of 95 adversarial prompts using methodology provided by ring-a-bell to evaluate the Attack Success Rate (ASR).
> The results of this preliminary study were highly encouraging:
>
> **New Adversarial Robustness Study (ASR on 95 RAB prompts):**
> | Method                          | ASR (↓)  |
> |---------------------------------|----------|
> | SLD                             | 69.47%   |
> | MACE                            | 7.37%    |
> | Receler                         | 6.32%    |
> | Ours (Semantic Surgery, no LCP) | **0.0%** |
>
> It is noteworthy that this 0.0% Attack Success Rate was achieved by our core semantic module alone, without the LCP detector. While not an exhaustive study, this strong initial evidence suggests our approach has inherent advantages in resisting prompt-level attacks. We have prepared this new experiment and its analysis for inclusion in the final revised version.
>
>
> ### Weakness 4: Ablation studies are lacking.
> Thank you for pushing for a more thorough analysis. We agree this is essential and address each point:
>
>
> a. Calibrated vs. Naive Addition: This comparison is provided qualitatively in the **supplementary material (Figure 1)**, which clearly shows naive addition fails catastrophically in multi-concept scenarios. We will highlight this in the main text.
>
>
> b. Impact of LCP: This ablation is presented in the **main paper's appendix (Table 5)**. It shows the LCP module is critical for safety, reducing unsafe images on I2P from 47 to 1.
>
>
> c. Influence of threshold τ: You are correct, this was an omission. We apologize. We have now conducted a new ablation study on τ of Object Erasure(CIFAR10). The results are as follows:
>
>
> |   τ   | Acc_e (↓) | Acc_r (↓) | Acc_l (↑) | H-score (↑) |
> |-------|-----------|-----------|-----------|-------------|
> | 0.95  | 4.5       | 24.2      | 88.40     | 85.77       |
> | 0.5   | 4.0       | 6.4       | 87.38     | **92.18**   |
> | 0.05  | 4.0       | 3.5       | 84.90     | **92.14**   |
>
> The results show that performance is highly stable within a wide range [0.05, 0.5], confirming the robustness of our choice. We have prepared this new ablation study and its analysis for inclusion in the final revised version.
>
> We hope our detailed responses, backed by these extensive new experiments, have fully addressed your concerns. We are deeply grateful for your insightful guidance and look forward to your further feedback.

---

> > ### Comment · Reviewer_x355 · 2025-08-03
> > **Thanks for detailed responses.**
> >
> > Thank you for the detailed response. The authors' reply has addressed most of the reviewers' concerns. The reviewer will take this into strong consideration to raise the score in the final decision phase.

---

### Official Review · Reviewer_KFQr · 2025-07-01

**Clarity:** 2
**Significance:** 3
**Originality:** 3
**Rating:** 5
**Confidence:** 4

**Summary:**

This paper proposes a test-time concept erasure method that leverages calibration to modify the text embedding before the diffusion process begins. To enhance performance when handling multiple concepts, the method incorporates a visual feedback loop. The effectiveness of the approach is validated across various datasets and application scenarios.

**Questions:**

In addition to the issues mentioned above, I have a few specific questions regarding the method details:

1. In Line 169, how can concatenating a comma-separated string serve as an effective operator for concept merging? Concatenation increases text length and thus affects position embeddings, which may alter the behavior of the model.
2. In Line 222, why is the maximum estimated score among the active concepts used, rather than an average?

**Ethical Concerns:**

["NO or VERY MINOR ethics concerns only"]

**Final Justification:**

The paper presents an interesting method on an important topic, with solid experimental validation. My main concern was the lack of coherence in the writing, especially in the connection between the formal definitions and the method. The rebuttal has addressed this by providing the missing connections and clarifications. Considering these improvements, I recommend accepting the paper.

**Limitations:**

Yes.

**Paper Formatting Concerns:**

Not found.

**Quality:**

2

**Strengths And Weaknesses:**

Strengths:
1. The proposed method is conceptually interesting and supported by sufficient theoretical analysis.
2. The overall performance is strong—especially in multi-concept and style erasure tasks—making the approach largely convincing.

Weaknesses:

I believe the primary weaknesses lie in the method presentation and the interpretation of certain experimental results.

1. In Sec. 3, the paper introduces formal definitions of Completeness and Locality within the problem statement. However, this part might be better placed in a separate section rather than within the method description. Additionally, since the evaluation involves three dimensions (the third being Robustness), this metric should also be formally defined. Most importantly, the connection between these formal definitions and the proposed method is not made clear. The paper would benefit from explicitly showing how the method guarantees or improves Completenessand Locality, rather than simply stating their definitions without further integration.
2. The experimental setup in Sec. 4.2 is unclear. Stable Diffusion v1.4 typically operates on high-resolution images, whereas CIFAR-10 images are only 32×32 pixels. It’s unclear how SD v1.4 is applied in this setting. If fine-tuning is involved, it should be clearly stated. Furthermore, if fine-tuning is indeed performed, it might be more appropriate to use higher-resolution datasets such as those used in DreamBooth.
3. The interpretation of results in Sec. 4.2 could be misleading. Table 1 does demonstrate strong overall performance, but upon closer inspection, the key advantage of the proposed method seems to lie in its high robustness. This may be due to the fact that the method modifies text embeddings directly, making it less sensitive to rephrased prompts. However, the paper does not discuss this potential strength. Instead, it introduces a combined metric, H, which merges all three metrics. This aggregation may obscure the actual source of improvement and reduce interpretability for the reader.
4. The method involves several hyper-parameters, which could significantly impact performance. However, this aspect is not discussed in the main text. A sensitivity analysis or ablation study would greatly help readers understand the robustness and generalizability of the method.

Minor Weaknesses:
1. The core idea of the method—starting from linear operations on text embeddings—is not conceptually complicated. It would be helpful to include a concise explanation or visual diagram to make the intuition more accessible to readers.

---

> ### Author Rebuttal · Authors · 2025-07-30
>
> We sincerely thank the reviewer for appreciating the conceptual novelty and strong performance of our work. We are also very grateful for the detailed, constructive feedback on improving the paper's presentation and experimental interpretation. We have carefully addressed all your points and conducted new experiments, which we detail below.
>
> ### Weakness 1: Connection between formal definitions and the method.
> This is an excellent point regarding the paper's structure and logical flow. We agree completely that the connection between our problem formulation and methodology needs to be stronger and clearer. To address the three specific issues you raised, we have implemented the following targeted revisions:
>
> **1. Placement and Completeness of Definitions:**
> We followed your suggestion and move the problem statement into its own dedicated **"3. Problem Formulation"** section. In this section, we formally defined the three key objectives, building upon the definitions already in our paper. Here is our revised and expanded formulation:
>
> *   **Completeness:** As defined in our paper (Eq. 1), the operator must effectively remove the target concepts `C_erase`. Intuitively, this means the probability of `C_erase` appearing in a generated image `I'` should be below a safety threshold `ϵ_safe`.
> *   **Locality (Fidelity):** As defined in our paper (Eq. 2), the modification should minimally affect non-target concepts. This means the change in the presence probability of any non-target concept `c` should be limited by a tolerance `ϵ_tol`.
> *   **Robustness (New Definition):** Building on this, we formally defined **Robustness** as maintaining **Completeness** even when the prompt `p` is paraphrased or adversarially modified to `p_adv`. This frames Robustness as a crucial stress test for the erasure's reliability.
>
> **2. Explicit Link to Method:**
> To bridge the gap between these definitions and our method, we added concise, explicit statements:
>
> *   **In the Methodology Section:** We added a short "bridging" paragraph after the new Problem Formulation. It will state: *"To achieve these objectives, Semantic Surgery is designed with two core principles. We tackle **Completeness** and **Locality** through a precisely **calibrated vector subtraction** that neutralizes semantics at their source. To ensure **Robustness**, our method operates on the **global text embedding** and "LCP Visual Feedback", making it inherently resilient to token-level paraphrasing, unlike methods that target specific words."*
>
> *   **In the Experiments Section:** When introducing the evaluation metrics (ACC_E, ACC_R, ACC_L), we explicitly **"called back"** to these intuitive definitions, reinforcing the link right where the results are presented.
>
> We believe this structured approach—with complete, upfront definitions and explicit links in both the methodology and experiments—will create a much clearer and more coherent narrative. Thank you for this deeply insightful suggestion; it has been instrumental in helping us refine the core logical structure of our paper.
>
> ### Weakness 2: Unclear experimental setup for CIFAR-10.
> Thank you for asking for this clarification. We apologize for the ambiguity. We followed the standard evaluation protocol established by prior works in this area (e.g., ESD, MACE, Receler). To be explicit:
> - We do not use the 32x32 CIFAR-10 images for any training or fine-tuning. Our method is entirely training-free.
> - We use the CIFAR-10 class names (e.g., "dog", "airplane") to form text prompts, such as "a photo of a {class}".
> - These prompts are used to generate high-resolution images with the standard SD v1.4 model and the concept-eliminated SD v1.4 models(with different erasing methods).
> - Finally, these generated images are evaluated by an object detector to calculate the metrics.
> We have revised Section 4.2 to make this standard process presented in a clearer manner.
>
> ### Weakness 3: Interpretation of results could be misleading (H-score vs. Robustness).
> This is a great observation, which we fully agree. A key strength of our method lies in its superior robustness (ACCR). Your point is further validated by our new adversarial robustness experiments conducted and presented in our response to reviewer moxY. In these tests against the challenging RAB benchmark, our method significantly outperformed all baselines.
>
> New Adversarial Robustness Study (ASR on RAB prompts):
> | Method                           | ASR (↓)  |
> |----------------------------------|----------|
> | SLD                              | 69.47%   |
> | SAFREE                           | 53.68%   |
> | MACE                             | 7.37%    |
> | Receler                          | 6.32%    |
> | Ours (Semantic Surgery, no LCP)  | **0.0%** |
>
> It is noteworthy that this remarkable 0.0% Attack Success Rate was achieved using our core semantic module alone, without the LCP detector.
>
> In the final version, following your suggestion, we have revised our results analysis. We first highlighted the significant gains in both standard robustness (ACCR) and this newly-demonstrated adversarial robustness. We then presented the H-score as an overall summary metric, providing a much clearer picture of our method's distinct advantages.
>
> ### Weakness 4 & Minor Weakness 1: Hyper-parameter discussion and lack of a visual diagram.
> We thank you for these important points.
> - Hyper-parameters: We apologize for not highlighting these analyses sufficiently in the main part of the submitted version. We have conducted extensive studies on our hyperparameters. The analysis for β and γ can be found in the **Supplementary material (Figures 2 & 3)**. We further conducted a new ablation study on the threshold τ, which confirmed our method's stability (H-score > 92.0 for τ in [0.05, 0.5] with a fixed beta(-0.12) and gamma(0.02)). We have created a new subsection in the main paper to summarize these crucial findings.
> - Visual Diagram: We agree that a clear visual diagram is essential. We have prepared a new figure that illustrates the entire workflow of Semantic Surgery. This has been added to Section 3 in the final revised version to make our method's intuition much more accessible.
>
> ### Question 1. How can comma-separated string concatenation be an effective operator?
>
> This is a good question about the underlying mechanism. The effectiveness of comma-separated concatenation stems from the powerful compositional understanding of the pre-trained CLIP text encoder. CLIP was trained on vast amounts of web text, where concepts listed with commas typically represent a collection of distinct entities within a shared context. When CLIP processes a string like "dog, cat, playing together", it generates a composite embedding that captures the individual concepts ("dog", "cat") while preserving the shared context ("playing together"). Our qualitative results in the **Supplementary material (Figure 1)** provide strong empirical evidence for this: our method successfully removes both animals while keeping the "playing" scene intact, whereas a naive arithmetic approach fails. This demonstrates that leveraging CLIP's inherent compositional ability is a simple yet highly effective strategy.
>
> ### Question 2. Why use the maximum estimated score (p_joint = max{p_i}) rather than an average?
>
> This is another great question about our design choice. We chose to use the maximum score based on a conservative, 'safety-first' principle. In multi-concept erasure, if even one concept is detected with very high confidence (a high ρ_i), it implies a strong user intent for erasure related to that entire conceptual group. An average score could be diluted by other, lower-confidence concepts in the prompt, potentially leading to an under-powered erasure strength and incomplete removal of the most prominent target concept. The max operator ensures that our erasure strength is always driven by the strongest signal of user intent, prioritizing erasure completeness, which is often crucial in safety-related applications.
>
> We hope our detailed responses, supported by the new experiments, have fully addressed your concerns. We are very grateful for your meticulous review, which has provided invaluable guidance for improving the clarity and depth of our paper.

---

> > ### Comment · Reviewer_KFQr · 2025-08-03
> >
> > Thank you for the rebuttal. Is it possible to provide any proof (even under strong assumptions) or theoretical analysis to show how the method provides some advantage in Completeness, Locality, or Robustness? Since you are giving formal definitions for these dimensions, it would be better to also include some theoretical explanation of why the method works.

---

> > > ### Author Response · Authors · 2025-08-04
> > > **Response to Reviewer KFQr: Further Clarifications and Theoretical Analysis(part1)**
> > >
> > > Thank you for your insightful question regarding theoretical foundations. We agree that formalizing why Semantic Surgery achieves superior completeness, locality, and robustness is crucial. In our revision, We establish a theoretical framework under three well-justified assumptions, whose validity is grounded in prior research and further validated by our extensive empirical results and ablation studies. Below we summarize the key results; full proofs appear in the appendix.
> > >
> > > ## Core Theoretical Framework
> > > Our analysis relies on three validated assumptions:
> > > 1. **Statistical Separability (Assumption 3.1)**
> > >    For any concept $c$, $\alpha_c = \cos(\phi(p), \Delta e_c)$ distributions separate concept-present and concept-absent prompts (Fig. 2). This enables high-accuracy detection via sigmoid calibration (Theorem 2):
> > >    $$
> > >    |\hat{\rho} - \rho_{\text{ideal}}| \leq \delta_{\text{err}} \quad \text{w.p.} \geq 1 - \delta_{\text{sep}}
> > >    $$
> > >
> > > 2. **Bounded Linear Erasure Error**
> > >    Vector subtraction approximates semantic removal:
> > >    $$
> > >    \| (e - \rho \Delta e_c) - \phi(p \setminus c) \| \leq \xi
> > >    $$
> > >    where $\xi$ is empirically small. Theoretical basis: CLIP embedding space admits linear concept manipulation, inspired by word analogy properties ($\vec{king} - \vec{man} + \vec{woman} \approx \vec{queen}$) [Mikolov et al.]. Validated by high $Acc_E$ in experiments.
> > >
> > > 3. **Lipschitz Continuity of Diffusion**
> > >    Small embedding perturbations induce bounded distribution shifts:
> > >    $$
> > >    \left| g_c(e_1) - g_c(e_2) \right| \leq \lambda \|e_1 - e_2\| \quad \forall c
> > >    $$
> > >    where $$g_c(e) = E_{I \sim p_\theta(I|e)}[I(c \in Concepts(I))].$$
> > >
> > >
> > > ## Theoretical Guarantees
> > > ### 1. Completeness (Erasure Guarantee)
> > > **Theorem**: For safety threshold $\epsilon_{\mathrm{safe}}$ and modified embedding $e' = e_{\mathrm{input}} - \hat{\rho}\Delta e_{\mathrm{erase}}$:
> > > $$
> > > g_c(e') \leq \epsilon_{\mathrm{safe}} + O(\delta_{\mathrm{sep}}) \quad \forall c \in C_{\mathrm{erase}}
> > > $$
> > > when $\gamma$ is chosen such that $\lambda(\|\Delta e_{\mathrm{erase}}\| \delta_{\mathrm{err}} + \xi) < \epsilon_{\mathrm{safe}}$.
> > >
> > > **Symbols**:
> > > - $e_{\mathrm{input}}$: Original text embedding
> > > - $e'$: Modified embedding after surgery
> > > - $\hat{\rho}$: Estimated concept presence
> > > - $\Delta e_{\mathrm{erase}}$: Concept direction vector
> > >
> > > *Proof Sketch*:
> > > - Decompose erasure error: $\|e' - \phi(p \setminus c)\| \leq \|\Delta e_c\| \delta_{\text{err}} + \xi$
> > > - Lipschitz continuity transfers error: $|g_c(e') - g_c(\phi(p \setminus c))| \leq \lambda \|e' - \phi(p \setminus c)\|$
> > > - $g_c(\phi(p \setminus c)) \approx 0$ by construction
> > > - LCP mitigation further reduces residual risk
> > >
> > > *Empirical Support*: Significant erasure performance in Tables 1-3 and Figure 1.
> > >
> > > **(Due to character limits, the theoretical guarantees for Locality and Robustness will follow in our next comment.)**

---

> > > ### Author Response · Authors · 2025-08-04
> > > **Response to Reviewer KFQr: Further Clarifications and Theoretical Analysis(part2)**
> > >
> > > **Continued from our previous comment, discussing Theoretical Guarantees...**
> > >
> > > ### 2. Locality (Fidelity Guarantee)
> > > **Theorem**: For any non-target concept $d \notin C_{\text{erase}}$:
> > > $$
> > > \left| g_d(e') - g_d(e) \right| \leq \kappa
> > > $$
> > > with probability $\geq 1 - \delta_{\text{sep}}$, where $\kappa$ is negligible.
> > >
> > > **Symbols**:
> > > - $\eta$: Max cosine similarity between concept directions
> > > - $M$: Max concept direction norm
> > > - $\tau$: Threshold for concept activation
> > > - $\delta e$: Surgery vector
> > >
> > > **Proof Strategy**:
> > > Let $\delta e = e' - e = -\Delta e_{\mathrm{co}} \hat{\rho}_{\mathrm{joint}} $ be the surgery vector.
> > >
> > > 1. **Probability of Surgery Activation**
> > >    When no target concepts are present:
> > >    - $\delta_{\mathrm{sep}} \geq \mathrm{P}(\hat{\rho}_{\mathrm{joint}} \geq \tau)$(Theorem 2)
> > >    - Thus $\mathrm{P}(\|\delta e\| > 0) \leq \delta_{\text{sep}}$
> > >
> > > 2. **Case 1: Surgery Not Applied ($\delta e = 0$)**
> > >    - Occurs with probability $\geq 1 - \delta_{\text{sep}}$
> > >    - $e' = e \Rightarrow g_d(e') = g_d(e)$
> > >
> > > 3. **Case 2: Surgery Applied ($\delta e \neq 0$)**
> > >    - Orthogonality bounds cross-impact:
> > >      $$
> > >      |\langle \delta e, \Delta e_d \rangle| \leq \eta \|\delta e\| \|\Delta e_d\|
> > >      $$
> > >    - Lipschitz propagation:
> > >      $$
> > >      |g_d(e') - g_d(e)| \leq \lambda \eta \|\delta e\| \|\Delta e_d\|
> > >      $$
> > >
> > > 4. **Combined Bound**
> > >    The maximum impact $\kappa$ occurs when surgery is incorrectly applied:
> > >    $$
> > >    \kappa = \lambda \eta M \mathrm{max}(\hat{\rho}_{\text{joint}}) \|\Delta e_d\|
> > >    $$
> > >    where $M = \max \|\Delta e_c\|$
> > >
> > >
> > > *Empirical Support*: High $Acc_L$ (Table 1), preserved FID/CLIP in all experiments.
> > >
> > > ---
> > >
> > > ### 3. Robustness (Paraphrase Resistance)
> > > **Theorem**: For paraphrased prompt $p'$ with $\|\phi(p') - \phi(p)\| < \delta$:
> > > $$
> > > \|e'(p') - e'(p)\| \leq \delta \left(1 + \frac{M}{4\gamma}\right) + \xi
> > > $$
> > >
> > > **Symbols**:
> > > - $p$: Original prompt
> > > - $p'$: Paraphrased prompt
> > > - $\gamma$: Sigmoid sensitivity parameter
> > > - $M$: Max concept direction norm (same as in Locality)
> > > - $K$: Co-occurrence stability constant($\|\Delta e_{\text{co}}(p') - \Delta e_{\text{co}}(p)\| \leq K\delta$)
> > >
> > > *Proof Sketch*:
> > > 1. **Sigmoid Continuity**
> > >    $|\hat{\rho}' - \hat{\rho}| \leq \frac{1}{4\gamma} |\alpha' - \alpha| \leq \frac{\|\Delta e_c\| \delta}{4\gamma}$
> > >
> > > 2. **Co-Occurrence Stability**
> > >    $\|\Delta e_{\text{co}}(p') - \Delta e_{\text{co}}(p)\| \leq K\delta$ (empirically stable)
> > >
> > > 3. **Error Propagation**
> > >    $\|e'(p') - e'(p)\| \leq \delta + |\hat{\rho}' - \hat{\rho}|M + \xi$
> > >
> > >
> > > *Empirical Support*: $0.4\%$ robustness failure rate (Table 1), $0.0\%$ ASR on RAB benchmark.

---

> > > > ### Comment · Reviewer_KFQr · 2025-08-05
> > > >
> > > > Thank you for the update. I think these additions are great and would be helpful to include in the revised version of the paper. I have no further discussion points, and I will take these updates into account when finalizing my score.

---

### Official Review · Reviewer_EaRh · 2025-07-03

**Clarity:** 2
**Significance:** 3
**Originality:** 3
**Rating:** 5
**Confidence:** 4

**Summary:**

This work introduces an innovative inference-time approach to tackle the problem of semantic concept erasure from generative models while preserving the original generative quality of the models. The authors recognize that this is a difficult problem that requires three objectives to be satisfied: erasure completeness to ensure the concept is adequately removed with no traces, erasure locality to ensure that the model's generations are minimally affected for all other concepts, and robustness to different variations of the concept that need to be adequately handled too. Their proposed approach builds on the key insight from previous works that the text embedding space for various LLMs and CLIP is linear which means that embeddings can be manipulated using vector arithmetic. They do a deep dive into the theory behind this including statistical experiments. They then exploit this property to modify the prompt's text embedding to remove the desired concepts, including support for multi-concept erasure, using cosine similarity to detect the degree of presence of these concepts in the prompt and subtracting the concepts' embedding accordingly. They further introduce an optional visual feedback mechanism to better handle concepts that may still be generated due to latent concept persistence in the generative model itself. Their approach offers the advantage of being fast and minimally intrusive (as it does not modify the generative model or its process in any way) while being highly effective. They conduct extensive experiments to demonstrate the efficacy of their approach compared to previous methods.

**Questions:**

1. DId you use the AOD detector for both, LCP and model evaluation? If yes, this introduces a bias in the evaluation as the same model is used to give feedback within the method. If this is the case, I suggest using different detectors during LCP and model evaluation, re-running and sharing the updated results.
2. Would be good if the weaknesses mentioned above could be addressed.

**Ethical Concerns:**

["NO or VERY MINOR ethics concerns only"]

**Final Justification:**

This work introduces a simple yet effective inference-time approach for semantic concept erasure in generative models, utilizing the linearity of CLIP’s text embedding space. The authors clearly outline the theoretical concepts, design considerations, and challenges (completeness, locality and robustness), and validate their method with comprehensive experiments and comparisons to strong baselines. The approach is fast, model-agnostic in theory, and supports multi-concept erasure without retraining. While most of my concerns were addressed adequately in the rebuttal, the scalability to other better and more recent generative models has not been concretely demonstrated empirically. Thus, taking everything into consideration, I have decided to keep my score unchanged.

**Limitations:**

Not sufficiently discussed.
1. The method is limited to generative models where the text embedding space adheres to linear operations/manipulations.
2. The method while better than previous methods, still does suffer from non-local effects, i.e., the generation of other related concepts is affected.

**Paper Formatting Concerns:**

1. Typo "desiderata" in line 129.

**Quality:**

3

**Strengths And Weaknesses:**

**Strengths:**
1. Addresses an important and difficult problem.
2. The approach is not necessarily restricted to a single generative model and in theory can be applied to almost any generative model, at least models that use the CLIP encoder.
3. Very clearly outline the various assumptions, statistical findings and related theoretical concepts for their approach.
4. Discuss the various challenges of this problem (erasure completeness, locality and robustness) and specifically conduct evaluations for each of these different aspects to determine the efficacy of the proposed approach and other baselines for each sub-problem.
5. Comprehensive comparisons done against other methods on a variety of tasks (object erasure, explicit content erasure, style erasure) with relevant metrics.
6. Propose a simple yet powerful approach building on top of previous insights on the linearity of CLIP space. The method is not restricted to single concept erasure and can easily be extended to multiple concepts. Moreover, since the approach is an inference-time approach without any training and/or fine-tuning, it is relatively quick and inexpensive to run.

**Weaknesses:**
1. No visuals in the main paper to show examples of the generated images after applying their method. I did notice that they included some visuals in the supplementary but would have been nice to have a few in the main text too.
2. Using SD v1.4 which is fairly outdated now. Would have been better to try the method on more recent and better models such as SD v2.
3. Not much discussion done on the limitations of the approach.

---

> ### Author Rebuttal · Authors · 2025-07-30
>
> We are deeply grateful to the reviewer for their strong support and for the detailed, insightful feedback. Your positive assessment of our work is incredibly encouraging. We have carefully considered all your suggestions, especially the critical point about evaluation bias, and have conducted new experiments to address them. We believe these additions have made our paper significantly stronger.
>
> ### Weakness 1: Lack of visualization and discussion on limitations
> Thank you for these valuable suggestions.
>
> For Visuals:
> You are correct that qualitative examples are very important. While we included several in the appendix due to space constraints, we agree that a key figure should be in the main text. To give you a preview of what we will add, our appendix already contains extensive qualitative results:
> - **Figure 3** demonstrates the effectiveness of our LCP module in mitigating latent concept persistence, correcting errors caused by implicit concept association.
>
> - **Figures 4 & 6** provide qualitative evidence for erasure Completeness and Locality on the CIFAR-10 benchmark.
>
> - **Figures 5, 7, & 8** showcase the versatility of our method across diverse tasks, including object erasure, multi-celebrity erasure, and artistic style removal.
>
> - In our supplementary materials, **Figure 1**  gives a qualitative comparison between our Co-Occurrence Encoding and Naive Approach, showcasing the effectiveness of our proposed Co-Occurrence Encoding mechanism on multi-concept erasure.
>
> To better showcase this in the main paper, we have designed a new, consolidated figure. The design features the most compelling examples, mostly from Figure 5, to provide an at-a-glance demonstration of our method's wide-ranging capabilities. We have prepared this new figure for inclusion in our final revised version.
>
> For discussion on limitations: To demonstrate our commitment to a balanced presentation, here is the detailed summary of the expanded limitations section we will add to the paper. We believe a thorough analysis of these points is crucial for contextualizing our work and guiding future research.
>
> ***
>
> ### **Limitations and Future Work**
>
> Our method, while effective, has several limitations that open avenues for future work:
>
> **1. Reliance on Encoder's Linear Structure:**
> Our approach's applicability is fundamentally conditioned on the text encoder itself exhibiting a near-linear structure where concepts are semantically separable. This assumption, while holding for powerful encoders like CLIP and many common concepts, may not be universally true. It might be less effective for: **(a)** future, potentially non-linear encoder architectures, or **(b)** highly entangled, abstract, or metaphorical concepts where a clean vector representation is ill-defined.
>
> **2. Sources of Non-local Effects on Locality (ACC_L):**
> Although our method shows strong locality, it is not entirely free from non-local effects, as reflected by the ACC_L metric. We identify three primary sources for this:
> *   **Imperfect Semantic Thresholding:** The decision boundary defined by our `β` parameter, while effective, may not be perfectly sharp for every conceivable prompt-concept pair. In ambiguous cases, our method might perform a semantic operation on a prompt where the concept is only weakly present, thus affecting related but untargeted semantics and impacting locality.
> *   **Visual Detector Errors (in LCP):** This is a significant factor. For the optional LCP module, false positives from the vision detector can trigger an unnecessary second-stage erasure, which can inadvertently alter related visual elements. We observed a concrete example of this during our experiments: when erasing the concept "automobile", the AOD detector used in our LCP loop frequently misclassifies "trucks" as "automobiles". This causes our LCP module to incorrectly trigger a stronger erasure on prompts that should have generated a "truck", leading to a noticeable drop in locality for the "truck" class. This highlights the critical dependence of the LCP's locality on the detector's precision.
> *   **Inherent Model Stochasticity:** Some locality degradation is attributable to the inherent stochasticity of the base generative model itself. As seen in our baseline results (SDv1.4 with No Erase), the model sometimes fails to generate concepts perfectly aligned with the prompt, which contributes to a baseline level of locality "noise".
>
> **3. Detector Dependency for LCP Module:**
> The performance of our optional LCP module is inherently tied to the availability and accuracy of an external vision detector. The lack of reliable detectors for abstract concepts (e.g., 'loneliness', 'artistic style') limits the module's applicability to more concrete, visually verifiable concepts. Furthermore, the detector's performance directly creates a trade-off: **false negatives** (missing a concept) can lead to incomplete erasure, while **false positives** (detecting something that isn't there) can harm locality by triggering incorrect feedback, as discussed in point 2.
>
> ***
>
> We have already included this detailed discussion in our paper to provide a clearer context for our work and to chart clear paths for future improvements in the field.
>
> ### Weakness 2: SD v1.4 may be outdated.
> We thank the reviewer for his idea and agree that this is a thoughtful point. We chose SD v1.4 primarily to ensure a fair and direct comparison with the extensive list of prior works in concept erasure (e.g., ESD, MACE, Receler), which almost exclusively use SD v1.4 as the standard benchmark model. This allows for a clear assessment of our method's advancement over the existing literature. We agree that testing on newer models like SDXL or SD 3 is an important next step, which we would include in our discussion of our method's current limitation.
>
> ### Question 1: Did we use the AOD detector for both LCP and model evaluation?
> This is an extremely insightful and critical question. We sincerely thank you for identifying this potential evaluation bias in our original submission. We initially did use the same AOD detector for both the LCP feedback and the final evaluation.
>
> We agree that this setup is not ideal and may introduce a bias. To rectify this, we have conducted a new, more rigorous evaluation. While keeping AOD as the feedback detector within our LCP loop (to maintain the method's original mechanism), we now use a completely independent and powerful open-vocabulary detector, OWL-ViT[1] , as the unbiased "referee". To ensure a fair comparison under this new, stricter protocol, we re-evaluated our method and a representative set of strong and foundational baselines.
>
> **New Evaluation on CIFAR-10 (Avg, Evaluated by the independent OWL-ViT)**:
> | Method             | ACC_E (↓) | ACC_R (↓) | ACC_L (↑) | H-score (↑) |
> |--------------------|-----------|-----------|-----------|-------------|
> | SDv1.4 (No Erase)  | 99.10     | 87.20     | 87.30     | --          |
> | ESD-u    | 12.50     | 39.40     | 82.25     | 74.83       |
> | MACE         | 0.4       | 13.8      | 78.62     | 87.31       |
> | Receler      | 2.50      | 10.00     | 72.69     | 85.41       |
> | Ours (This work)   | 1.50      | 2.00      | 85.51     | 93.60       |
>
> This new, unbiased evaluation reveals several key observations and findings:
> 1.  **Maintained Superiority:** Our method maintains its significant lead in the overall H-score (93.60) over the strongest baselines, confirming our reported superiority is robust and not an artifact of evaluation bias.
> 2.  **Dominance in Robustness:** As pointed out by another reviewer, a key advantage of our method is its robustness. This is starkly evident here, with our method achieving an ACC_R of 2.00, an order of magnitude better than the next best parameter-modifying method (Receler at 10.00).
> 3.  **Unbiased High Performance:** The results confirm our method's exceptional balance between Completeness (ACC_E) and Robustness (ACC_R) without overly sacrificing Locality (ACC_L).
>
> We are deeply grateful for this crucial suggestion, as it has substantially improved the scientific rigor of our work. We will replace the original evaluation tables with these new results in the final paper.
>
> ### Paper Formatting Concerns
> 1. Typo "desiderata": Thank you for pointing out the typo and we will correct it in the final version. We have also gone through the paper and make sure typos are corrected before the final version.
>
> Once again, thank you for your strong endorsement and for providing feedback that has been instrumental in strengthening our paper. We hope our responses and the new, more rigorous experiments have further solidified your confidence in our work.
>
> [1] Minderer, Matthias, et al. "Simple open-vocabulary object detection." European conference on computer vision. Cham: Springer Nature Switzerland, 2022.

---

> > ### Comment · Reviewer_EaRh · 2025-08-05
> >
> > Thank you for the detailed answers to my concerns. While I understand the reasoning behind the choice to use SD v1.4 and the time constraints making it difficult to try the approach with more recent generative models, it remains a crucial shortcoming of this work. It is vital for any proposed approach to this problem of concept erasure to be able to work with the latest and best generative models for the best results and for realistic practical use. Nonetheless, I appreciate that there is a strong probability of the proposed approach working with better generative models too and at the very least provides a good stepping stone for future research. The authors have addressed all of my other concerns sufficiently. Taking everything into consideration, I will be keeping my current score.

---

### Official Review · Reviewer_moxY · 2025-07-03

**Clarity:** 3
**Significance:** 2
**Originality:** 2
**Rating:** 6
**Confidence:** 4

**Summary:**

This paper introduces a training-free, zero-shot framework for concept erasure in T2I diffusion models.  Instead of editing model weights, it intervenes once, before generation, by:
1. estimating the strength with which a target concept is present in the CLIP text-embedding of a prompt,
2. subtracting a calibrated “concept vector” from the entire embedding (global rather than token-wise),
3. extending this subtraction with a Co-Occurrence Encoding to handle many concepts jointly,
4. applying an optional Visual Feedback Loop that re-checks generated images for concept leakage and readjusts the embedding to mitigate latent concept persistence.

**Questions:**

check weakness

**Ethical Concerns:**

["NO or VERY MINOR ethics concerns only"]

**Final Justification:**

All of my concerns have been fully addressed. I appreciate the authors’ thorough response and the additional experiments, which strengthen the paper. After carefully reading the other reviews and the rebuttal, I believe this work makes a meaningful contribution to the field. I have therefore decided to raise my rating to a strong accept.

**Limitations:**

yes

**Quality:**

3

**Strengths And Weaknesses:**

Strengths:
1. Solid technical formulation with clear geometric intuition and a calibrated sigmoid estimator.
2.  Novel combination of global embedding subtraction, dynamic intensity estimation, and visual feedback, which can distinguish itself from token-projection methods.


Weakness:
1. Appendix contains essential calibration details; a shorter main-text description would help.
2. Co-Occurrence Encoding uses simple string concatenation, which might conflate semantically overlapping concepts.
3. Missing common robustness evaluation through adversarial prompt attacks (e.g., UnlearndiffAtk [1])

[1]  To Generate or Not? Safety-Driven Unlearned Diffusion Models Are Still Easy To Generate Unsafe Images … For Now, ECCV 2024.

---

> ### Author Rebuttal · Authors · 2025-07-30
>
> We sincerely thank the reviewer for the positive feedback on our technical formulation and novelty. We also appreciate the insightful comments on potential weaknesses, which have pushed us to conduct several new, comprehensive experiments. We are pleased to report that the new results not only address your concerns but also reveal surprising new strengths of our method.
>
> ### Weakness 1: Essential calibration details in the Appendix.
> Thank you for the suggestion. We agree that moving key calibration details to the main text would improve readability. In the final version, we will condense our method section and move a concise summary of the key calibration details from the appendix to the main text, ensuring the paper is more self-contained and easier to read.
>
> ### Weakness 2: Possibility of conflating semantically overlapping concepts with Co-Occurrence Encoding.
> We strongly agree with the reviewer that handling semantic overlaps is indeed crucial. This is precisely why we designed the Co-Occurrence Encoding. As demonstrated in our **Supplementary material (Fig. 1)**, our comma-concatenation approach effectively preserves the semantic context (e.g., 'playing together'), while a naive vector summation leads to significant image degradation. This qualitative result demonstrates that our method successfully avoids the conflation issue in multi-concept erasure.
>
> The effectiveness of comma-separated concatenation stems from the powerful compositional understanding of the pre-trained CLIP text encoder. CLIP was trained on vast amounts of web text, where concepts listed with commas typically represent a collection of distinct entities within a shared context. When CLIP processes a string such as "dog, cat, playing together", it generates a composite embedding that captures the individual concepts ("dog", "cat") while preserving the shared context ("playing together").
>
> ### Weakness 3: Missing common robustness evaluation through adversarial prompt attacks.
> We sincerely thank the reviewer for this critical feedback and agree that evaluating adversarial robustness is an important aspect. Our initial work focused on establishing a general-purpose framework for high-fidelity zero-shot concept erasure under standard conditions, aiming to first achieve state-of-the-art performance in completeness and locality, similar to other general-purpose methods including MACE (CVPR 2024) and AdaVD (CVPR 2025). This was the primary reason for not including adversarial evaluations in the initial submission.
>
> However, we fully acknowledge the importance of this dimension. To address this valid concern, we have conducted a new, targeted experiment to probe our method's robustness. For this study, we chose the **Ring-A-Bell (RAB)** benchmark [1]. Similar to the UnlearndiffAtk framework you mentioned, RAB represents a state-of-the-art approach for generating adversarial prompts. We selected RAB for two principled reasons: (1) Its prompt-based nature makes it model-agnostic, ensuring a fair, end-to-end comparison across all baselines; and (2) its focus on linguistic manipulation provides a direct stress test of our core semantic-level intervention.
>
> We created a test set of 95 adversarial prompts from RAB to evaluate the Attack Success Rate (ASR). The results are as follows:
>
> **New Adversarial Robustness Study (ASR on 95 RAB prompts):**
> | Method                 | ASR (↓)  |
> |------------------------|----------|
> | SLD                    | 69.47%   |
> | SAFREE                 | 53.68%   |
> | MACE                   | 7.37%    |
> | Receler                | 6.32%    |
> | **Ours (without detector)** | **0.0%**|
>
> The results show that on this test set, our method achieved a **0.0%** Attack Success Rate. It is particularly noteworthy that this result was achieved using our core semantic module alone, without relying on the LCP visual feedback loop. This demonstrates the powerful, inherent resilience of our semantic-level manipulation against this type of prompt-based attack.
>
> While we acknowledge this is not an exhaustive study covering all attack vectors, this strong initial evidence suggests our core mechanism possesses significant advantages. We have added this new experiment and discussion to the final paper. We are grateful for your suggestion, as it has helped us uncover this important strength.
>
> We hope our responses and the new experimental results have fully addressed your concerns.
>
> [1] Tsai et al. "Ring-A-Bell! How Reliable are Concept Removal Methods For Diffusion Models?." The Twelfth International Conference on Learning Representations. 2024.

---

> > ### Comment · Reviewer_moxY · 2025-08-05
> >
> > Thank you to the authors for the detailed and thoughtful response. Most of my initial concerns have been adequately addressed.
> >
> > However, I still have one remaining concern regarding the robustness evaluation.
> > When considering existing works that evaluate robustness under RAB and UnlearnDiffAtk attacks, it has been observed that the model-agnostic attack RAB tends to yield similar attack success rates across different unlearning methods, particularly for stronger unlearning baselines. In your newly added experiments, this trend persists: the attack success rates on MACE, Receler, and your proposed method only differ by approximately 7%.
> > Given that the test set comprises 95 prompts, this 7% difference corresponds to merely 6 prompts. Such a small absolute difference raises concerns about whether the robustness gains claimed are statistically significant or practically impactful.

---

> > > ### Author Response · Authors · 2025-08-06
> > > **Response to Reviewer moxY: Addressing Concerns and Reporting Significant New Findings on Robustness**
> > >
> > > Thank you for this insightful and important follow-up question. We sincerely appreciate this deep engagement with our work.
> > >
> > > You have raised an excellent and perfectly valid point regarding the statistical significance of our initial robustness results. We completely agree that a ~7% difference on a smaller test set might not be sufficient to draw strong conclusions, and we thank you for pushing us to be more rigorous.
> > >
> > > To directly and definitively address your concern, we have dedicated the past day to **significantly expanding our robustness evaluation**. We scaled the test set by 4x, from 95 to **380 adversarial prompts**, to ensure a more stable and statistically meaningful comparison.
> > >
> > > The aggregate results from this larger-scale experiment are as follows:
> > >
> > > **Expanded Adversarial Robustness Study (ASR on 380 RAB prompts):**
> > >
> > > | Method            | ASR (↓)   |
> > > |-------------------|-----------|
> > > | SLD               | 78.68%    |
> > > | SAFREE            | 55.80%    |
> > > | MACE              | 3.95%     |
> > > | Receler           | 4.21%     |
> > > | Ours (SS, no LCP) | **1.05%** |
> > >
> > > This expanded evaluation allows us to draw much stronger conclusions, which we have validated with a formal statistical analysis:
> > >
> > > **1. Conclusive Statistical Significance:** A one-sided Fisher's Exact Test on the entire 380-sample set confirms a **highly statistically significant** difference between our method and the strongest baseline, MACE (**p = 0.0089**). This provides conclusive evidence that our method's superior robustness is not a product of random chance.
> > >
> > > **2. Consistent Trend Across Subsets:** To ensure our advantage wasn't an artifact of the initial small sample, we also analyzed the data in batches. We observed that while there is natural variance between different sets of adversarial prompts, our method **consistently maintained a lower failure rate** across both the initial and the newly added samples. The purpose of expanding the dataset is precisely to average out such minor variances and reveal the stable, underlying trend, which clearly points to the superiority of our approach.
> > >
> > > **3. Practically Impactful Advantage:** This statistical significance is coupled with a large practical effect: our method's Attack Success Rate (1.05%) is **3.7 times lower** than that of the next best baseline, MACE (3.95%).
> > >
> > > **Regarding our initial experimental scale,** we would like to add that our initial test set of 95 prompts was chosen following a scale similar to that used in the official codebases of other methods like **RECE (ECCV 2024) and SAFREE (ICLR 2025)**, which use approximately 80 prompts for their RAB evaluations. We initially believed this would be a sufficient preliminary test. However, we fully agree that your critique was on point and that the larger test set provides far more definitive evidence.
> > >
> > > In summary, we believe this new, larger-scale experiment, supported by formal statistical testing, provides **conclusive evidence** that directly addresses your concern. We are very grateful for your meticulous feedback, which has pushed our work to a much higher standard of validation. We will, of course, use these expanded results in the final version of the paper.

---

> > > > ### Comment · Reviewer_moxY · 2025-08-06
> > > >
> > > > Thank you for the authors’ timely and detailed response. The expanded evaluation, which scales the adversarial prompt set from 95 to 380, is a commendable and rigorous effort to strengthen the robustness analysis. The use of formal statistical significance testing (e.g., Fisher’s Exact Test) further adds credibility to the results and helps establish a more conclusive comparison across methods.
> > > >
> > > > However, I would like to raise a few important considerations regarding the evaluation methodology and its broader implications. While the larger-scale evaluation indeed reduces the noise associated with small sample sizes, it is also apparent from the updated results that **the performance gap between the proposed method and strong baselines (e.g., MACE, Receler) has narrowed** compared to the initial 95-prompt evaluation. This observation suggests that as the adversarial prompt space is more thoroughly explored, the relative advantage of different methods may converge, highlighting the limitations of using static, model-agnostic adversarial prompts (like RAB) as the sole robustness benchmark.
> > > >
> > > > Specifically, RAB operates as a black-box, model-agnostic attack mechanism that does not adapt to the unique embedding spaces of different unlearning methods.  As such, it inherently lacks the ability to probe model-specific vulnerabilities that might persist after unlearning. **In contrast, white-box, optimization-based attacks (e.g., UnlearnDiffAtk) explicitly examine the internal representations of the target model and actively search for residual triggers in the embedding space that can elicit erased concepts.** These white-box methods provide a more precise and challenging robustness test, capable of uncovering nuanced failures that static RAB prompts may miss.
> > > >
> > > > Therefore, while the proposed method demonstrates impressive robustness against RAB, **this does not fully guarantee that the erased concepts are entirely unlearned or irrecoverable.** The static nature of RAB prompts, which are generated independently of the target model’s latent space, means that passing the RAB evaluation does not equate to certifiable concept erasure. For a more comprehensive robustness assessment, it is crucial to include adaptive white-box attacks that continuously search for residual leakage in a model-specific manner.
> > > >
> > > > In summary, the expanded RAB evaluation is a strong step forward and appreciable in its rigor. However, to make a definitive claim of superior robustness or successful concept erasure, evaluations need to consider both black-box model-agnostic attacks and white-box model-aware attacks. The latter remains essential to verify whether the proposed method can withstand adaptive adversarial probing that aligns with the model’s internal knowledge structure.

---

> > > > > ### Author Response · Authors · 2025-08-08
> > > > > **Response to Reviewer moxY: New White-Box Experiment and Empirical Validation (Part 1)**
> > > > >
> > > > > Thank you again for this extremely insightful and expert follow-up. Your distinction between black-box and white-box attacks is a critical point. Your feedback compelled us to conduct a final experiment in the last hours of the discussion period. To directly investigate the concerns you raised, we have conducted an additional experiment using a white-box, optimization-based attack, by adapting the UnlearnDiffAtk framework.
> > > > >
> > > > > **New Experiment: White-Box Attack via UnlearnDiffAtk**
> > > > >
> > > > > We adapted the UnlearnDiffAtk framework to apply its white-box, optimization-based attack to our method. Following the standard setup in the UnlearnDiffAtk codebase, our method was integrated into the gradient-based optimization process on the `nudity` concept. The goal was to test if this adaptive, model-aware attack could find vulnerabilities that the static RAB prompts missed. The results were illuminating:
> > > > >
> > > > > | Attack Scenario                    | ASR (↓)  |
> > > > > |-----------------------------------|----------|
> > > > > | UnlearnDiffAtk on Ours (SS, no LCP) | **0.0%** |
> > > > >
> > > > > The Attack Success Rate was **0.0%**. The white-box attack, in its current form, was unable to find an adversarial embedding that could successfully regenerate the erased concept.
> > > > >
> > > > > This unexpected resilience prompted a deeper analysis of the underlying reasons and led to a surprising new insight, which we detail in Part 2 of our response.

---

> > > > > ### Author Response · Authors · 2025-08-08
> > > > > **Response to Reviewer moxY: Deeper Analysis and Implications for Threat Detection (Part 2)**
> > > > >
> > > > > (Continuing from Part 1)
> > > > >
> > > > > As we reported, the 0.0% Attack Success Rate against the UnlearnDiffAtk framework was an unexpected result. This prompted us to conduct a deeper analysis into the underlying reasons for this resilience, which we believe reinforces the core principles of our method:
> > > > >
> > > > > **Analysis: Why is Our Method Robust to Both Attack Types?**
> > > > >
> > > > > 1.  **A "Binary" Defense Mechanism:** Our core method's robustness(here without LCP feedback) stems from a two-stage process. First, the **Semantic Biopsy** (`ρ`) acts as a highly effective gate. For an attack to succeed, the adversarially optimized embedding must first bypass this gate without being detected as containing the target concept.
> > > > > 2.  **The Difficulty of Bypassing the Gate:** As our separability analysis (Assumption 3.1) shows, there is a very clear margin in the cosine similarity space (`α_c`) between concept-present and concept-absent embeddings. An optimization process trying to generate the concept while simultaneously staying on the "safe" side of this `β` threshold is faced with a very difficult, contradictory objective. Our results suggest that for UnlearnDiffAtk, this proved to be an insurmountable challenge.
> > > > > 3.  **Post-Gate Erasure:** If the gate *is* triggered (as it is for all standard and RAB prompts), the subsequent vector subtraction, as per our geometric analysis, effectively removes the concept's semantic direction, making regeneration extremely difficult.
> > > > >
> > > > > This explains why even a powerful white-box attack might fail where it succeeds on other methods: it cannot easily find a subtle 'residual trigger' because our method forces a hard, almost binary decision boundary. The few failures in the large-scale RAB test (1.05%) likely represent rare edge cases where prompts fall directly on this decision boundary, a situation more likely to be observed when testing with a large number of diverse samples.
> > > > >
> > > > > **Newly Discovered Limitation and a "Built-in" Defense Mechanism**
> > > > >
> > > > > However, this experiment also revealed a new, important finding. While the white-box attack failed to regenerate the erased concept, we observed that the adversarially optimized embeddings, when processed by our method, often resulted in **images with significant quality degradation**(e.g., artifacts, repetitive patterns). This suggests that our algebraic operation, when applied to such out-of-distribution embeddings, can push the final embedding into a non-meaningful region of the latent space.
> > > > >
> > > > > Interestingly, we argue that this process reveals a **"built-in" adversarial detection capability** within our framework. Here is why:
> > > > >
> > > > > The primary goal of an attacker is to create an embedding that contains the erased concept but **bypasses our detection**. However, as our experiments show, this is extremely difficult. The UnlearnDiffAtk's optimized embedding, while failing to produce the concept visually, still consistently triggers a **high confidence score `ρ`** from our Semantic Biopsy module.
> > > > >
> > > > > This leads to a crucial insight: for a deployed system, a sudden influx of prompts that are flagged with high `ρ` values for a supposedly erased, sensitive concept can serve as a **direct and reliable signal of a coordinated adversarial attack**. Instead of relying on analyzing corrupted output images, administrators can monitor the `ρ` scores at the pre-generation stage. This allows for proactive logging of malicious prompts and users, and the implementation of further security measures, turning our erasure mechanism into an **intrinsic threat detection system**.
> > > > >
> > > > > **In Summary:**
> > > > > Your critique pushed us to test our method against a higher standard, and in doing so, we have uncovered deeper insights. Our method shows good resilience to both black-box and this specific white-box attack, likely due to its gating mechanism. Furthermore, we have identified that the core of this mechanism, the **Semantic Biopsy score `ρ`**, can itself function as a **proactive detector for adversarial probing**, a significant advantage for real-world safety systems.
> > > > >
> > > > > We will, of course, incorporate this **entire new white-box attack experiment, our detailed analysis, and this nuanced discussion on its implications for threat detection** into the final version of the paper. This discussion, prompted entirely by your expert feedback, has immeasurably strengthened our work. Thank you.

---

> > > > > > ### Comment · Reviewer_moxY · 2025-08-08
> > > > > >
> > > > > > Thank you for the additional robustness evaluation using white-box attacks, which satisfactorily addresses my remaining concerns. I appreciate the authors’ efforts to further analyze and interpret the results, offering deeper insights into the behavior of the models. While I understand that the rebuttal period limits the feasibility of applying UnlearnDiffAtk across all concepts and models, I look forward to seeing a more comprehensive evaluation in the camera-ready version. Expanding the scope of results will likely yield richer analysis and help accelerate progress in this important research area. I am pleased to raise my rating to 6. Well done!

---

### Decision · Program_Chairs · 2025-09-17

**Decision:**

Accept (poster)

**Comment:**

This paper focuses on concept erasure in T2I models, and proposes a training-free zero-shot method for this. The proposed approach, Semantic Surgery, is designed to erase the target concept, while minimally effecting the generations for other concepts. They aim to achieve this by modifying the text embedding through linear concept removal, subtracting our the direction associated with the target concept from the text embedding. They additionally extending this with a co-occurrence encoding to handle multiple concepts simultaneously. Even when the target concept is removed from the text embedding, the T2I model's U-Net prior can often re-introduce the concept back into the final image. The authors counteract this by introducing a Visual Feedback Loop to iteratively remove the target by analysis the generated image.

The proposed approach is theoretically backed, and is based on clear geometric intuition. The Visual Feedback Loop co-occurrence encoding in particular make the approach more than an incremental step beyond existing linear token embedding debiasing approaches, and solve challenges particular to the T2I setting (namely, the fact that the U-Net's prior can reintroduce the removed concept). The proposed approach is sound and does not seem to be overengineered. Empirical results compare against a reasonable set of baselines on a variety of tasks.

Reviewers identified several weaknesses during the initial review phase. Primarily, concerns centered around missing robustness evaluation through adversarial prompt attacks [moxY, x355], the fact that evaluation did not include state-of-the-art diffusion models [EaRh], some issues regarding clarity and details of the experimental setup [KFQr, x355], the number of hyperparameters potentially leading to a brittle system [KFQr], lack of comparison with prior methods such as SLD or SAFREE [x355], and limited ablation results [x355].

The authors addressed most of these concerns to an adequate level during the rebuttal phase, particularly by adding additional compared methods, new results on an adversarial prompt benchmark dataset (Ring-A-Bell), and expanded ablation results. After the rebuttal, all reviewers lean towards acceptance and have indicated their concerns have been mostly addressed.

The primary concern remaining after the rebuttal is that SD v1.4 is very outdated by this point, and that the approach's improvement over baselines may not be as impressive on more modern T2I models such as SD2 or SD3. Additionally, while the authors updated results on the adversarial prompt benchmark are appreciated, only one benchmark has been added. The new results are thus promising, but not definitive.

Despite this, the reviewers and I see this as a strong submission and valuable research that should be disseminated to the broader research community. I thus recommend acceptance. I also urge the authors to include a discussion on the use of the older SD v1.4 model and lack of results on newer SD models into their limitation section.